# Adaptive Energy Amplification for Robust Time Series Forecasting

## Abstract

Deep learning models for time series forecasting often exhibit a spectral bias, prioritizing high-energy, low-frequency components while underfitting predictive but low-energy, high-frequency signals. Existing efforts attempt to correct this by amplifying high-frequency components but suffer from indiscriminate amplification, enhancing both meaningful signals and task-irrelevant noise, which destabilizes training and impairs generalization. To address this, we propose **AEA** (**A**daptive **E**nergy **A**mplification), a novel framework that reframes the problem as one of adaptive signal enhancement. AEA introduces two synergistic innovations: (1) a **Spectral Mirroring** mechanism that constructs a phase-preserving, low-frequency surrogate to guide targeted, distortion-free amplification of high-frequency signals; and (2) a lightweight **Differential Embedding** module that operates in a latent space to adaptively suppress common-mode noise. By decoupling signal amplification from noise suppression, AEA selectively enhances only informative features. Extensive experiments show that our model-agnostic framework consistently improves the forecasting performance of state-of-the-art backbones in both long-term and short-term forecasting tasks, while significantly enhancing training stability and generalization. The code repository is available at `https://anonymous.4open.science/r/AEA-685E/`.

## 1 Introduction

Time series forecasting (TSF) is critical in various real-world applications, including traffic flow prediction (Wu et al., 2020), energy management (Zhou et al., 2021), weather forecasting (Liang et al., 2023), financial investment (Oreshkin et al., 2020), human healthcare (Qiu et al., 2024), *etc*. Recent deep learning-based methods, which have powerful nonlinear modeling capabilities to learn complex patterns and feature representations, achieving remarkable performance on TSF, such as Convolutional-based (Wu et al., 2023; donghao & wang xue, 2024), Transformer-based (Nie et al., 2023; Liu et al., 2024a), and MLP-based methods (Zeng et al., 2023; Wang et al., 2024).

Despite these advances, such models exhibit a fundamental spectral bias: they consistently prioritize high-energy, low-frequency components while overlooking subtle yet predictive high-frequency signals (Xu et al., 2024; Yi et al., 2024). As shown in Figure 1a, masking low-frequency components causes a drastic drop in performance, while masking high-frequency components only marginally impacts performance, revealing the models' over-reliance on low-frequency information with limited capability for modeling high-frequency signals. This learning pathology originates from the model's optimization bias on low-frequency components with high energy. According to *Parseval's Theorem* (Lathi & Green, 1998; Yi et al., 2023), the energy is equivalent between the time and frequency domains. In most real-world time series data, low-frequency components possess substantially higher amplitudes than their high-frequency counterparts, meaning energy is concentrated in the low-frequency part of the spectrum. As a result, the predictive loss landscape becomes dominated by errors from these low-frequency components with high amplitude. This skews the optimization process, compelling the learning algorithm to primarily allocate model capacity toward fitting these dominant, low-frequency signals, while the informative yet low-energy high-frequency details are consequently underfitted (Liu et al., 2023; Piao et al., 2024; Fei et al., 2025).

To address the issue, recent efforts have focused on amplifying the energy of high-frequency components to recalibrate their influence during model optimization. These methods can be broadly

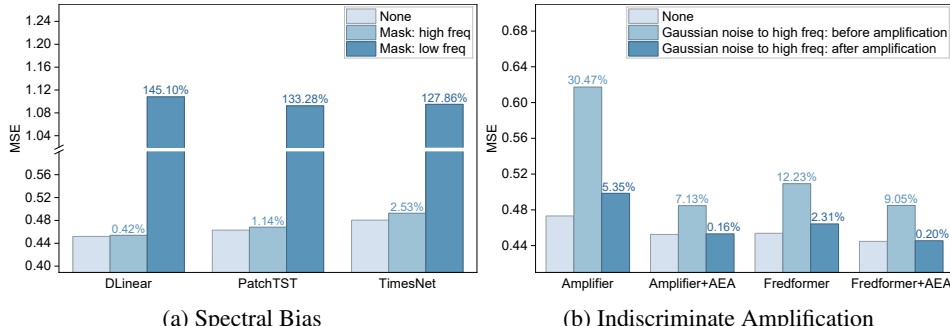

(a) Spectral Bias          (b) Indiscriminate Amplification

Figure 1: The average degradation in forecasting performance (Values denote relative increase in MSE (%) compared to the "None" baseline) during the inference stage on ETTh1. (a) When the lower *v.s.* higher 50% of frequency bands are masked (set to zero), the significantly smaller performance drop after high-frequency masking confirms the base model's reliance on low-frequency information and its insensitivity to high-frequency components. (b) Comparing the vanilla amplification method with our enhanced version (+AEA) when artificially injecting Gaussian noise of the same intensity into high frequencies. The results show that the vanilla methods' performance degrades drastically, proving that they are susceptible to noise. In contrast, our method successfully suppresses noise, resulting in significantly enhanced robustness. We present more details in Appendix B.1.

categorized into two main strategies: indirect and direct enhancement. Specifically, the **indirect enhancement** approaches mitigate energy disparity through normalization. For instance, Fredformer (Piao et al., 2024) implements frequency-wise local normalization, which segments the spectrum and normalizes each sub-band individually to eliminate amplitude disparity. On the other hand, the **direct enhancement** strategy, conversely, explicitly modifies the spectral energy distribution. Amplifier (Fei et al., 2025) is one representative work, whose key innovation is spectrum flipping. This technique inverts the spectrum to leverage high-energy signals as a template for boosting low-energy signals.

However, despite their different mechanisms, these approaches share a fundamental flaw: their amplification is indiscriminate. High-frequency bands inherently contain a mixture of predictive signals (*e.g.*, subtle seasonal variations and trends) and task-irrelevant noise (*e.g.*, sensor artifacts and background noise) (Eldele et al., 2024; Kou et al., 2025; Yi et al., 2025). By uniformly elevating the energy across the high-frequency bands, existing methods inevitably amplify noise alongside the valuable signals. This indiscriminate enhancement introduces spectral disturbances that destabilize the optimization process and ultimately impair the model's generalization performance. As empirically demonstrated in Figure 1b, when noise is injected into the high-frequency bands, both indirect and direct enhancement methods suffer a significant performance degradation, underscoring the negative impact of their indiscriminate amplification and revealing an inherent inability to distinguish between informative signals and spurious noise.

To address this limitation, we argue that *the key to unlocking the potential of high-frequency signals lies not in indiscriminate amplification, but in adaptive enhancement*. We introduce AEA (**A**daptive **E**nergy **A**mplification), a novel framework that fundamentally reframes the problem by simultaneously amplifying signals and suppressing noise. AEA achieves this through two synergistic innovations designed to provide a more principled and reasonable energy amplification. First, (1) **Spectral Mirroring** addresses the amplification itself by leveraging the typically cleaner, high signal-to-noise ratio of the low-frequency spectrum. It constructs a phase-preserving surrogate from these reliable low-frequency components to serve as a structured template, guiding a targeted amplification of high-frequency signals without introducing spectral distortion. Second, to explicitly tackle noise, (2) **Differential Embedding** operates in a learned latent space to identify and filter out common-mode noise, which indiscriminate methods inadvertently amplify. By integrating these two mechanisms, AEA ensures that only the informative, discriminative features within the high-frequency bands are selectively enhanced, thereby resolving the core issue of indiscriminate amplification by separating the targeted enhancement of predictive signals from the active suppression of noise.

In summary, our contributions can be highlighted as followings:

- We systematically identify the problem of "indiscriminate amplification" in forecasting models against spectral bias, establishing a novel connection between targeted energy amplification and adaptive noise suppression.

- We propose AEA, a model-agnostic framework that employs spectral mirroring for distortion-free amplification and differential embedding for adaptive noise suppression, seamlessly integrating with various forecasting backbones.

- We empirically demonstrate that AEA consistently improves accuracy, stability, and generalization across multiple benchmark datasets and state-of-the-art backbones in both short-term and long-term forecasting, offering a robust new paradigm for frequency-aware time series forecasting.

## 2 RELATED WORK

### 2.1 TIME SERIES FORECASTING MODELS

Traditional time series forecasting methods such as ARIMA (Zhang, 2003) and Prophet (Taylor & Letham, 2018; Triebe et al., 2021) are effective at capturing trend and seasonal components in time series (Cleveland et al., 1990; Ahmed et al., 2010; Wen et al., 2020; Zeng et al., 2023; Stitsyuk & Choi, 2025). With the continuous growth in data availability, deep learning methods have brought revolutionary advances to the field, introducing more complex and efficient models (Torres et al., 2021; Lim & Zohren, 2021). Convolutional Neural Networks (CNNs) (Bai et al., 2018; Wan et al., 2019; Sen et al., 2019; Liu et al., 2022a; Wu et al., 2023) have been widely adopted to capture local temporal dependencies, while Recurrent Neural Networks (RNNs) (Rangapuram et al., 2018; Smyl, 2020; Salinas et al., 2020; Hewamalage et al., 2021), although proficient at processing sequential information, often struggle with long-sequence modeling. Transformer-based models (Zhou et al., 2021; Wu et al., 2021; Liu et al., 2022b; Zhang & Yan, 2022; Nie et al., 2023; Liu et al., 2024a; Wen et al., 2023; Tang & Matteson, 2021; Zhou et al., 2022b; Liu et al., 2021; Feng et al., 2024), typically equipped with self-attention mechanisms (Vaswani et al., 2017), excel at capturing long-range dependencies, albeit at considerable computational cost. Recently, linear models (Oreshkin et al., 2020; Zhang et al., 2022; Das et al., 2023) such as DLinear (Zeng et al., 2023) and TSMixer (Chen et al., 2023) have gained popularity due to their simplicity and strong performance in long-term forecasting, though they may underperform on highly non-linear and complex patterns. Furthermore, multi-periodicity analysis (Benaouda et al., 2006; Percival & Walden, 2000; Wu et al., 2023; Wang et al., 2022; Chen et al., 2024; Yi et al., 2023; Zhou et al., 2022a) continues to play an essential role in the preprocessing stages of advanced modeling pipelines.

### 2.2 FREQUENCY DOMAIN METHODS IN TIME SERIES FORECASTING

Recent studies have increasingly leveraged frequency-domain techniques to enhance the accuracy and efficiency of time series forecasting (Yi et al., 2025). Prominent examples include FEDformer (Zhou et al., 2022b), which accelerates attention via frequency-domain low-rank approximation; FreTS (Yi et al., 2023), which integrates global frequency properties into an efficient MLP architecture; and FITS (Xu et al., 2024), which employs frequency interpolation as an effective low-pass filter. A common characteristic of these approaches is their tendency to prioritize high-energy, low-frequency components, a design choice that aligns with the natural energy distribution of many real-world time series. However, this emphasis may lead to insufficient use of subtle yet predictive high-frequency signals, which often carry critical short-term variations and anomaly patterns. The challenge of effectively balancing frequency components without amplifying noise has thus emerged as a key issue in frequency-aware forecasting. Recent efforts have attempted to address this spectral imbalance. Fredformer (Piao et al., 2024) mitigates frequency bias in Transformers by promoting more balanced feature learning across bands, yet its architecture-specific design limits generalizability. Amplifier (Fei et al., 2025) directly elevates high-frequency energy to match low-frequency levels, aiming to equalize gradient scales across the spectrum. However, such uniform amplification risks enhancing high-frequency noise alongside signals. In contrast to these end-to-end architectures, our proposed AEA is designed as a model-agnostic plugin that decouples signal amplification from noise suppression. By combining targeted energy amplification and adaptive noise suppression, AEA achieves more nuanced enhancement while maintaining robustness and efficiency across diverse forecasting backbones.

## 3 PRELIMINARIES

**Time Series Forecasting**. Formally, let $X = [\boldsymbol{x}_1, \dots \boldsymbol{x}_T] \in \mathbb{R}^{T \times C}$ be a time series, where $T$ is the length of historical data. $\boldsymbol{x}_t \in \mathbb{R}^C$ represents the observation at time $t$. $C$ denotes the number of variates (*i.e.,* channels). The objective is to construct a predictive model $f$ that estimates the future values of the series, $Y = [\hat{\boldsymbol{x}}_{T+1}, \dots, \hat{\boldsymbol{x}}_{T+H}] \in \mathbb{R}^{H \times C}$, where $H$ is the forecasting horizon.

**Real Fast Fourier Transform**. Given a real-valued sequence $x[n]$ of length $N$, we employ the Real Fast Fourier Transform (rFFT) (Sorensen et al., 1987) to efficiently convert it into the frequency domain, and transform it back using the inverse rFFT (irFFT). The rFFT/irFFT exploits the conjugate symmetry of real-valued inputs, reducing the computational complexity from $O(N^2)$ to $O(N \log N)$ while compressing the output to $N/2 + 1$ complex-valued frequency components. The resulting spectrum $\mathcal{X} \in \mathbb{C}^{N/2+1}$ contains both magnitude and phase information:

$$A[k] = |\mathcal{X}[k]|, \quad \theta[k] = \angle \mathcal{X}[k] \tag{1}$$

where $A[k]$ represents amplitude and $\theta[k]$ phase at frequency $\omega_k = 2\pi k/N$. We provide more details of the Fourier Transform in Appendix A.1.

## 4 PROPOSED METHOD

### 4.1 OVERALL ARCHITECTURE

We propose the Adaptive Energy Amplification (AEA) framework to address the limitations of indiscriminate amplification in existing frequency-domain forecasting methods. As illustrated in Figure 2, AEA operates primarily in the frequency domain and consists of two core innovations: (1) a Spectral Mirroring module that performs targeted amplification of high-frequency signals via a phase-preserving surrogate spectrum, and (2) a Differential Embedding module that suppresses common-mode noise in a latent space to enhance discriminative features. To ensure spectral consistency, we incorporate an Energy Predictor that aligns the predictions with the original data distribution. The entire framework is model-agnostic and seamlessly integrates with various forecasting backbones. We present the pseudo-code in Algorithm 1.

### 4.2 SPECTRAL MIRRORING

The Spectral Mirroring achieves targeted amplification by constructing a phase-preserving surrogate from reliable low-frequency components. This focus on phase coherence distinguishes our approach from conventional spectral manipulation methods, which often introduce phase distortions that degrade signal reconstruction. Our method explicitly maintains phase relationships through a mixing strategy, enabling distortion-free enhancement of informative high-frequency components.

To enhance attention to low-energy, high-frequency components as well as high-energy, low-frequency components, we reverse the entire spectrum to create a structured surrogate (Fei et al., 2025). For an input spectrum $\mathcal{X}[k]$ with $k = 0, 1, \dots, F - 1$ (where $F = \lfloor T/2 \rfloor + 1$), the reversed spectrum is obtained by:

$$\mathcal{X}_{\text{reverse}}[k] = \mathcal{X}[F - 1 - k]. \tag{2}$$

This operation inverts the natural energy distribution, allowing the typically dominant low-frequency components to guide the amplification of subtle high-frequency signals.

To enable adaptive control over the amplification process, we introduce a learnable scaling matrix $M \in \mathbb{R}^{F \times C}$ that operates on the reversed spectrum:

$$\mathcal{X}_{\text{scaled}}[k, c] = \mathcal{X}_{\text{reverse}}[k, c] \cdot M[k, c], \quad \text{for } k = 0, 1, \dots, F - 1; c = 0, 1, \dots, C - 1. \tag{3}$$

This matrix allows the model to learn appropriate amplification factors for each frequency component and channel independently.

The key to avoiding distortion lies in how we mix the original and mirrored spectra. A simple linear combination of amplitudes and phases would likely result in destructive interference (Demirel & Holz, 2025). Instead, we employ a *phase mixing* strategy that minimizes disruptive phase discontinuities.

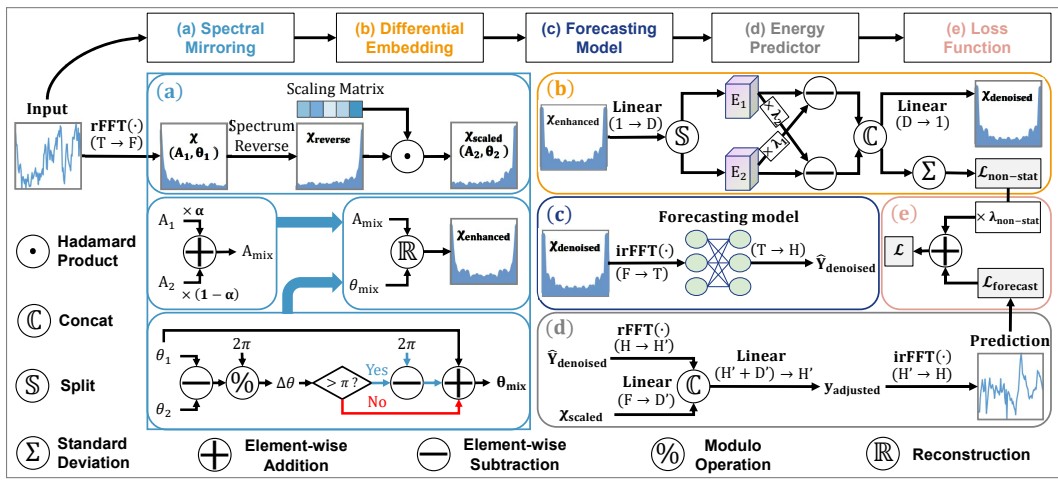

Figure 2: An illustration of the proposed AEA framework. The input time series is first transformed into the frequency domain. The framework consists of five components: (a) The Spectral Mirroring module (Section 4.2) that reverses the spectrum to adaptively amplify high-frequency signals without distortion through a learned scaling matrix and phase-preserving mixing. (b) The Differential Embedding module (Section 4.3) that projects the enhanced spectrum into a latent space to suppress common-mode noise via a differential operation, and yields the non-stationarity loss to stabilize learning. (c) The denoised spectrum is converted back to the time domain via irFFT for the base model to forecast. (d) The Energy Predictor module (Section 4.4) that aligns the output of the base model with the original data's spectral properties. (e) The optimization of the entire framework by a combined loss function (Section 4.5) comprising the forecast error and the non-stationarity loss.

The *phase mixing* involves calculating the circular difference between the original and scaled phases, adjusting it to the shortest angular path within $[-\pi, \pi]$, and then blending the phases accordingly:

$$\Delta\theta[k, c] = (\theta_1[k, c] - \theta_2[k, c]) \mod 2\pi, \tag{4}$$

$$\Delta\theta_{\text{adjusted}}[k, c] = \begin{cases} \Delta\theta[k, c] - 2\pi, & \text{if } \Delta\theta[k, c] > \pi, \\ \Delta\theta[k, c], & \text{otherwise,} \end{cases} \tag{5}$$

$$\theta_{\text{mix}}[k, c] = \theta_1[k, c] + \Delta\theta_{\text{adjusted}}[k, c], \tag{6}$$

where $\theta_1[k, c] = \angle(\mathcal{X}[k, c])$ and $\theta_2[k, c] = \angle(\mathcal{X}_{\text{scaled}}[k, c])$ represent the phase angles of the original and scaled spectra, respectively.

For amplitude combination, we employ a balanced mixing approach:

$$A_{\text{mix}}[k, c] = \alpha \cdot A_1[k, c] + (1 - \alpha) \cdot A_2[k, c], \tag{7}$$

where $A_1[k, c] = |\mathcal{X}[k, c]|$, $A_2[k, c] = |\mathcal{X}_{\text{scaled}}[k, c]|$, and $\alpha = 0.5$ provides equal weighting to both spectral representations. Ablation studies (refer to Appendix C.3) show that a fixed $\alpha = 0.5$ yields optimal performance. This balanced weighting stabilizes the initial enhancement, allowing subsequent adaptive components to focus on refining the signal.

The enhanced spectrum is then reconstructed as:

$$\mathcal{X}_{\text{enhanced}}[k, c] = A_{\text{mix}}[k, c] \cdot e^{j\theta_{\text{mix}}[k, c]}. \tag{8}$$

This process preserves temporal structure while amplifying informative high-frequency components, providing refined input for subsequent processing. The stability of the mirroring process is formally guaranteed by Theorem A.5, which ensures bounded deviation from the original signal characteristics throughout the enhancement process.

### 4.3 DIFFERENTIAL EMBEDDING WITH NON-STATIONARITY LOSS

The Differential Embedding module suppresses common-mode noise while preserving discriminative signals in the enhanced spectrum $\mathcal{X}_{\text{enhanced}}$. It projects the input into an embedding space, applies a

differential operation for noise suppression, and incorporates a regularization loss to stabilize training. The input spectrum is first projected into a complex-valued embedding space:

$$E_1 \| E_2 = W_e \cdot \mathcal{X}_{\text{enhanced}} + b_e, \tag{9}$$

where $W_e \in \mathbb{C}^{D \times 1}$ and $b_e \in \mathbb{C}^D$ are learnable parameters, and $D$ is the embedding dimension. $E_1 \in \mathbb{C}^{F \times C \times \frac{D}{2}}$ and $E_2 \in \mathbb{C}^{F \times C \times \frac{D}{2}}$ are the subspaces of the differential embedding.

Inspired by the principle of *differential attention* (Ye et al., 2025; Wang et al., 2025a), we apply a differential operation in the embedding space as follows:

$$E_1' = E_1 - \lambda_1 \cdot E_2, \quad E_2' = E_2 - \lambda_2 \cdot E_1, \tag{10}$$

where $\lambda_1$ and $\lambda_2$ are learnable scalars. These scalars are initialized with a random constant $\lambda_{\text{init}}$ and a Softplus function (Nair & Hinton, 2010) to ensure they remain positive throughout training:

$$\lambda_1 = \text{Softplus}(\lambda_{\text{init}}), \quad \lambda_2 = \text{Softplus}(\lambda_{\text{init}}). \tag{11}$$

The results are concatenated to form the denoised embedding:

$$E' = \text{Concat}(E_1', E_2'), \quad E' \in \mathbb{C}^{F \times C \times D}. \tag{12}$$

This operation is theoretically grounded in noise suppression (see Proposition 4.1). The denoised embedding is projected back to the frequency domain:

$$\mathcal{X}_{\text{denoised}} = W_p \cdot E' + b_p, \quad \mathcal{X}_{\text{denoised}} \in \mathbb{C}^{F \times C}, \tag{13}$$

where $W_p \in \mathbb{C}^{1 \times D}$ and $b_p \in \mathbb{C}^1$ are learnable parameters.

Theoretical analysis 4.7 shows that this differential operation reduces gradient bias from common-mode noise while preserving beneficial stochastic variance. To further enhance stability, we introduce a non-stationarity loss that penalizes excessive variability in embedding magnitudes across batches:

$$\mathcal{L}_{\text{non-stat}} = \sqrt{\text{Var}_{\mathbf{x} \sim \mathcal{B}}(|E'|)}, \tag{14}$$

where $\mathcal{B}$ represents the current batch of samples. This loss term serves as a regularizer that encourages the model to learn stable, stationary representations that are robust to batch-wise variations. The final denoised spectrum $\mathcal{X}_{\text{denoised}}$ is transformed back to the time domain via inverse rFFT for forecasting. The stability of Differential Embedding is also guaranteed by Theorem A.5, which ensures that the mirroring process does not introduce excessive signal distortion.

## 4.4 ENERGY PREDICTOR

The Spectral Mirroring and Differential Embedding alter the energy distribution of the input signal. While this enhancement improves the model's ability to capture high-frequency components, directly using the base model's predictions on this enhanced and denoised signal would therefore yield outputs with inconsistent energy characteristics (Liu et al., 2023). To ensure spectral consistency with the original data, our Energy Predictor learns a frequency-domain mapping. It takes the scaled historical spectrum $\mathcal{X}_{\text{scaled}}$ and the base model's preliminary prediction $\hat{Y}_{\text{denoised}}$ as inputs. First, $\hat{Y}_{\text{denoised}}$ is transformed to the frequency domain via rFFT to obtain $\mathcal{Y}_{\text{denoised}}$. Then, $\mathcal{X}_{\text{scaled}}$ is projected into a latent embedding to represent the historical context. This embedding is concatenated with $\mathcal{Y}_{\text{denoised}}$, and the combined representation is passed through a final linear projection. This step yields the adjusted spectrum $\mathcal{Y}_{\text{adjusted}}$, effectively aligning the prediction's spectral properties with the original data. These operations are defined precisely as follows:

$$\mathcal{Y}_{\text{denoised}} = \text{rFFT}(\hat{Y}_{\text{denoised}}), \qquad\qquad \mathcal{Y}_{\text{denoised}} \in \mathbb{C}^{H' \times C}, \tag{15}$$

$$\mathcal{E} = W_1 \cdot \mathcal{X}_{\text{scaled}} + b_1, \qquad\qquad \mathcal{E} \in \mathbb{C}^{D' \times C}, \tag{16}$$

$$\mathcal{Y}_{\text{adjusted}} = W_2 \cdot \text{Concat}(\mathcal{E}, \mathcal{Y}_{\text{denoised}}) + b_2, \qquad\qquad \mathcal{Y}_{\text{adjusted}} \in \mathbb{C}^{(D'+H') \times C}, \tag{17}$$

where $H' = \lfloor H/2 \rfloor + 1$, and $W_1 \in \mathbb{C}^{D' \times F}$, $W_2 \in \mathbb{C}^{H' \times (D'+H')}$, $b_1 \in \mathbb{C}^{D'}$, and $b_2 \in \mathbb{C}^{H'}$ are learnable parameters.

Finally, the adjusted spectrum is transformed back to the time domain to produce the final prediction:

$$\hat{Y} = \text{irFFT}(\mathcal{Y}_{\text{adjusted}}). \tag{18}$$

Proposition A.9 theoretically demonstrates that the Energy Predictor, by utilizing the high-frequency-enhanced historical signal as guidance, intelligently calibrates the distribution of the predicted signal back to the characteristics of the original data without undermining the existing high-frequency gains. This ensures spectral alignment through controlled modifications rather than destructive adjustments that could revert to low-frequency dominance. We present the theoretical bound in Proposition A.7.

## 4.5 Loss Function

We follow a multi-task optimization framework (Vandenhende et al., 2022) that simultaneously ensures accurate forecasting while maintaining representation stability and formulates the loss function $\mathcal{L}$ as:

$$\mathcal{L} = \mathcal{L}_{\text{forecast}} + \lambda_{\text{non-stat}} \cdot \mathcal{L}_{\text{non-stat}}, \tag{19}$$

where $\mathcal{L}_{\text{forecast}}$ is the Mean Squared Error (MSE) between predictions $\hat{Y}$ and ground truth $Y$, $\mathcal{L}_{\text{non-stat}}$ is the regularization term introduced in Equation 14, and $\lambda_{\text{non-stat}}$ is a balancing hyperparameter.

## 4.6 Computational Complexity Analysis

AEA is designed as a plug-and-play framework whose computational overhead is typically negligible compared to forecasting backbones, especially those with quadratic complexity. The complexity is dominated by FFT operations and linear projections across its modules. The rFFT/irFFT transformations require $O(T \log T)$ per channel. Spectral Mirroring performs element-wise operations in $O(T)$ time. Both Differential Embedding and Energy Predictor involve linear projections with complexity $O(F \cdot C \cdot D)$ or $O(F \cdot C \cdot D')$, where $F = \lfloor T/2 \rfloor + 1 \approx T$, $D$ and $D'$ are fixed (typically 64-128). Thus, the overall complexity of AEA is linear in both sequence length and number of channels, i.e., $O(T \cdot C)$. This is significantly more efficient than the quadratic complexity $O(T^2 \cdot C)$ of transformer backbones, making AEA a practical enhancement for real-world forecasting applications.

## 4.7 Theoretical Analysis

**Notation.** Let $e^{(1)}, e^{(2)} \in \mathbb{C}^{F \times C \times \frac{D}{2}}$ denote the two embedding subspaces from Equation 9, $\Theta$ the model parameters of loss $L$, $\lambda \in \mathbb{R}^+$ a learnable scaling parameter, $s_i$ the true signal component, $n_i^{(c)}$ the common-mode noise, and $\epsilon_i$ the independent stochastic noise.

**Proposition 4.1** (Adaptive Noise Suppression via Differential Embedding). *The differential embedding mechanism $e^{(\text{diff})} = e^{(1)} - \lambda e^{(2)}$ provides adaptive suppression of common-mode noise while preserving discriminative signals. The resulting gradient estimates $\hat{g} = g + \delta$ exhibit superior bias-variance trade-off:*

$$\mathbb{E}[\delta] = (1 - \lambda^*)\mathbf{b}_g, \quad Var(\delta) = (1 - \lambda^*)^2\sigma_c^2 + (1 + \lambda^{*2})\sigma_\epsilon^2 \tag{20}$$

*where $\lambda^*$ is the optimal value minimizing the training objective, $\mathbf{b}_g$ is the bias introduced by common-mode noise, $\delta$ is the gradient noise, and $\sigma_c^2$, $\sigma_\epsilon^2$ represent the gradient variances from common-mode and stochastic noise components, respectively.*

*Proof.* We begin by decomposing the embedding into signal and noise components under Assumption A.1, which is supported by previous studies (Ye et al., 2025; Wang et al., 2025a). This common-mode noise often stems from systematic biases present in the input data (*e.g.*, stop words in NLP, background regions in spatio-temporal data, or certain frequency components in the spectrum (Eldele et al., 2024)). The differential operation yields:

$$e^{(\text{diff})} = s^{(\text{diff})} + (1 - \lambda)n^{(c)} + (\epsilon^{(1)} - \lambda\epsilon^{(2)}) \tag{21}$$

Taking expectation over stochastic noise ($\mathbb{E}[\epsilon^{(1)}] = \mathbb{E}[\epsilon^{(2)}] = 0$):

$$\mathbb{E}[\delta] = (1 - \lambda)\mathbb{E}\left[\frac{\partial L}{\partial e^{(\text{diff})}} \cdot \frac{\partial n^{(c)}}{\partial \Theta}\right] = (1 - \lambda^*)\mathbf{b}_g \tag{22}$$

The variance analysis follows from the uncorrelatedness of noise components. The complete derivation, including detailed expectations, variance decompositions, and convergence guarantees, is

provided in Appendix A.4. This proposition validates our differential embedding by ensuring that: (i) common-mode noise amplified during spectral mirroring is effectively suppressed, (ii) the non-stationarity loss $\mathcal{L}_{\text{non-stat}}$ in Equation 14 stabilizes training by controlling gradient variance, and (iii) the adaptive parameter $\lambda^*$ optimally balances noise suppression against signal preservation throughout optimization. □

## 5 EXPERIMENT

To validate the effectiveness of the proposed AEA, we conduct extensive experiments on a variety of time series forecasting tasks, including both long-term and short-term forecasting.

### 5.1 EXPERIMENTAL SETUP

**Base models.** AEA is a model-agnostic framework that can be seamlessly integrated with arbitrary time series forecasting models to enhance their performance. To comprehensively evaluate its effectiveness, we select a broad range of state-of-the-art base models spanning both time-domain and frequency-domain paradigms. The time-domain models include DLinear (Zeng et al., 2023) (Linear), TimesNet (Wu et al., 2023) (CNN), and PatchTST (Nie et al., 2023) (Transformer), while the frequency-domain models comprise Amplifier (Fei et al., 2025), FreTS (Yi et al., 2023), and FredFormer (Piao et al., 2024). This comprehensive selection covers major architectural paradigms while specifically assessing AEA's value when integrated with specialized frequency-aware methods.

**Implementation details.** To ensure a fair and controlled comparison across all methods, we implement base models within a unified experimental framework based on the TimesNet codebase. This guarantees that all models were trained and evaluated under identical conditions, using the same data processing pipeline and training procedure. All experiments are conducted using PyTorch (Paszke et al., 2019) on a single NVIDIA RTX A100 80GB GPU. Experiment configurations and implementations are detailed in Appendix B.4.

### 5.2 LONG-TERM FORECASTING

**Setups.** We conduct long-term forecasting experiments on eight widely used real-world multivariate time series forecasting datasets, including ETT (ETTh1, ETTh2, ETTm1, and ETTm2), Electricity, Traffic, and Weather, which are utilized by Autoformer (Wu et al., 2021). Following the established evaluation protocol in TimesNet, we adopt Mean Squared Error (MSE) and Mean Absolute Error (MAE) as the primary evaluation metrics, as well as set the historical input length to 96, and forecasting horizons are evaluated at $\{96, 192, 336, 720\}$. To ensure a fair comparison, we consistently use the same experimental configuration as the original implementations. We split all forecasting datasets into training, validation, and test sets by the ratio of 6:2:2 for the ETT dataset and 7:1:2 for the other datasets. Details of metrics and datasets are in Appendix B.2 and Appendix B.3.

**Results.** The results in Table 1 demonstrate that the model enhanced with AEA outperforms the base model in general. Specifically, AEA improves forecasting performance in nearly 97% of cases for both MSE and MAE. A further analysis reveals that AEA brings an average performance gain of 3.817% for time-domain models and 2.393% for frequency-domain models, showing its complementary value even for specialized frequency-aware architectures. Remarkably, AEA achieves a substantial boost on TimesNet, with a significant reduction in MSE by 8.108% and MAE by 4.337%. The last column of the table quantifies the average percentage improvement in terms of MSE/MAE, at 3.105%, which underscores the consistent enhancement brought by AEA across all forecasting horizons and datasets.

### 5.3 SHORT-TERM FORECASTING

**Setups.** To comprehensively evaluate the generalizability of AEA and specifically investigate its capability in scenarios where high-frequency components are paramount, we extend our evaluation to short-term forecasting tasks. Experiments are conducted on the PeMS benchmark (Chen et al., 2001), a collection of high-dimensional traffic network datasets (PEMS03, PEMS04, PEMS07, PEMS08) where short-term, high-frequency variations are critical for accurate prediction. We follow

Table 1: Long-term forecasting performance comparison *w.r.t.* forecasting models with their counterparts enhanced by the AEA in terms of MSE and MAE, the lower the better. The forecasting horizons are {96, 192, 336, 720}. The better performance in each setting is shown in **bold**. 'Avg' denotes the average results of four forecasting horizons; The last column, 'IMP (%)', shows the average percentage of MSE/MAE improvement over all base models.

| Model | | DLinear | | + AEA | | PatchTST | | + AEA | | TimesNet | | + AEA | | Amplifier | | + AEA | | FreTS | | + AEA | | FredFormer | | + AEA | | IMP (%) |
|---|---|---|---|---|---|---|---|---|---|---|---|---|---|---|---|---|---|---|---|---|---|---|---|---|---|---|
| Metric | | MSE | MAE | MSE | MAE | MSE | MAE | MSE | MAE | MSE | MAE | MSE | MAE | MSE | MAE | MSE | MAE | MSE | MAE | MSE | MAE | MSE | MAE | MSE | MAE | |
| ETTh1 | 96 | 0.384 | 0.397 | 0.380 | 0.392 | 0.387 | 0.403 | 0.374 | 0.396 | 0.390 | 0.414 | 0.395 | 0.410 | 0.437 | 0.439 | 0.397 | 0.407 | 0.395 | 0.406 | 0.386 | 0.400 | 0.376 | 0.395 | 0.375 | 0.393 | 2.361 |
| | 192 | 0.433 | 0.427 | 0.432 | 0.422 | 0.455 | 0.444 | 0.428 | 0.431 | 0.519 | 0.486 | 0.441 | 0.436 | 0.451 | 0.436 | 0.440 | 0.428 | 0.454 | 0.443 | 0.441 | 0.436 | 0.438 | 0.427 | 0.434 | 0.426 | 3.798 |
| | 336 | 0.481 | 0.459 | 0.479 | 0.458 | 0.490 | 0.463 | 0.466 | 0.452 | 0.471 | 0.458 | 0.473 | 0.452 | 0.497 | 0.457 | 0.482 | 0.447 | 0.517 | 0.484 | 0.489 | 0.459 | 0.485 | 0.448 | 0.484 | 0.446 | 2.107 |
| | 720 | 0.509 | 0.506 | 0.495 | 0.489 | 0.510 | 0.496 | 0.479 | 0.478 | 0.543 | 0.511 | 0.490 | 0.476 | 0.508 | 0.484 | 0.492 | 0.475 | 0.589 | 0.536 | 0.529 | 0.513 | 0.506 | 0.479 | 0.479 | 0.441 | 5.428 |
| | Avg | 0.452 | 0.447 | 0.447 | 0.440 | 0.460 | 0.451 | 0.437 | 0.439 | 0.480 | 0.467 | 0.450 | 0.443 | 0.473 | 0.454 | 0.452 | 0.439 | 0.488 | 0.467 | 0.461 | 0.452 | 0.451 | 0.437 | 0.443 | 0.427 | 3.582 |
| ETTm1 | 96 | 0.344 | 0.371 | 0.332 | 0.368 | 0.327 | 0.366 | 0.327 | 0.365 | 0.357 | 0.389 | 0.338 | 0.377 | 0.323 | 0.363 | 0.321 | 0.361 | 0.339 | 0.374 | 0.323 | 0.366 | 0.339 | 0.369 | 0.327 | 0.349 | 2.510 |
| | 192 | 0.381 | 0.393 | 0.370 | 0.386 | 0.367 | 0.388 | 0.369 | 0.387 | 0.440 | 0.427 | 0.375 | 0.391 | 0.365 | 0.382 | 0.365 | 0.382 | 0.387 | 0.402 | 0.362 | 0.387 | 0.385 | 0.392 | 0.374 | 0.364 | 4.024 |
| | 336 | 0.416 | 0.418 | 0.402 | 0.408 | 0.405 | 0.417 | 0.406 | 0.410 | 0.410 | 0.421 | 0.410 | 0.414 | 0.397 | 0.403 | 0.393 | 0.401 | 0.420 | 0.426 | 0.394 | 0.413 | 0.418 | 0.413 | 0.394 | 0.394 | 2.467 |
| | 720 | 0.478 | 0.458 | 0.464 | 0.444 | 0.481 | 0.464 | 0.458 | 0.445 | 0.518 | 0.468 | 0.478 | 0.451 | 0.476 | 0.443 | 0.468 | 0.437 | 0.487 | 0.468 | 0.456 | 0.448 | 0.469 | 0.443 | 0.467 | 0.403 | 4.058 |
| | Avg | 0.405 | 0.410 | 0.392 | 0.402 | 0.395 | 0.409 | 0.390 | 0.402 | 0.431 | 0.427 | 0.400 | 0.408 | 0.390 | 0.398 | 0.387 | 0.395 | 0.408 | 0.417 | 0.384 | 0.403 | 0.403 | 0.404 | 0.391 | 0.427 | 3.337 |
| ETTh2 | 96 | 0.336 | 0.386 | 0.329 | 0.384 | 0.302 | 0.352 | 0.290 | 0.339 | 0.322 | 0.360 | 0.322 | 0.359 | 0.291 | 0.344 | 0.286 | 0.337 | 0.331 | 0.381 | 0.307 | 0.323 | 0.288 | 0.341 | 0.282 | 0.334 | 3.422 |
| | 192 | 0.452 | 0.459 | 0.450 | 0.457 | 0.416 | 0.424 | 0.375 | 0.392 | 0.428 | 0.427 | 0.419 | 0.417 | 0.370 | 0.396 | 0.369 | 0.389 | 0.509 | 0.492 | 0.441 | 0.453 | 0.371 | 0.389 | 0.361 | 0.381 | 4.210 |
| | 336 | 0.579 | 0.536 | 0.576 | 0.533 | 0.500 | 0.480 | 0.416 | 0.425 | 0.466 | 0.454 | 0.427 | 0.435 | 0.427 | 0.437 | 0.415 | 0.430 | 0.557 | 0.521 | 0.510 | 0.433 | 0.384 | 0.410 | 0.380 | 0.409 | 6.050 |
| | 720 | 0.784 | 0.638 | 0.795 | 0.641 | 0.482 | 0.478 | 0.421 | 0.439 | 0.443 | 0.440 | 0.418 | 0.438 | 0.439 | 0.452 | 0.424 | 0.443 | 0.819 | 0.654 | 0.793 | 0.631 | 0.416 | 0.435 | 0.409 | 0.433 | 2.860 |
| | Avg | 0.538 | 0.505 | 0.538 | 0.504 | 0.425 | 0.434 | 0.375 | 0.399 | 0.409 | 0.420 | 0.397 | 0.412 | 0.382 | 0.407 | 0.374 | 0.400 | 0.554 | 0.512 | 0.513 | 0.460 | 0.365 | 0.394 | 0.358 | 0.389 | 4.115 |
| ETTm2 | 96 | 0.188 | 0.283 | 0.189 | 0.286 | 0.181 | 0.266 | 0.182 | 0.265 | 0.187 | 0.264 | 0.175 | 0.257 | 0.178 | 0.260 | 0.177 | 0.260 | 0.183 | 0.271 | 0.172 | 0.263 | 0.178 | 0.261 | 0.176 | 0.260 | 1.505 |
| | 192 | 0.282 | 0.360 | 0.271 | 0.347 | 0.249 | 0.311 | 0.242 | 0.300 | 0.253 | 0.306 | 0.247 | 0.304 | 0.241 | 0.301 | 0.239 | 0.299 | 0.276 | 0.343 | 0.262 | 0.327 | 0.244 | 0.304 | 0.253 | 0.312 | 1.866 |
| | 336 | 0.360 | 0.411 | 0.372 | 0.420 | 0.311 | 0.352 | 0.305 | 0.342 | 0.323 | 0.349 | 0.316 | 0.345 | 0.299 | 0.340 | 0.297 | 0.339 | 0.336 | 0.384 | 0.329 | 0.348 | 0.306 | 0.344 | 0.293 | 0.335 | 1.800 |
| | 720 | 0.546 | 0.518 | 0.541 | 0.515 | 0.406 | 0.404 | 0.399 | 0.399 | 0.423 | 0.408 | 0.430 | 0.409 | 0.394 | 0.396 | 0.392 | 0.396 | 0.477 | 0.474 | 0.480 | 0.476 | 0.400 | 0.398 | 0.394 | 0.388 | 0.512 |
| | Avg | 0.344 | 0.393 | 0.343 | 0.392 | 0.287 | 0.333 | 0.282 | 0.326 | 0.296 | 0.332 | 0.292 | 0.329 | 0.278 | 0.324 | 0.279 | 0.323 | 0.318 | 0.368 | 0.311 | 0.354 | 0.282 | 0.326 | 0.279 | 0.324 | 1.246 |
| Electricity | 96 | 0.196 | 0.280 | 0.192 | 0.278 | 0.174 | 0.260 | 0.172 | 0.259 | 0.175 | 0.278 | 0.148 | 0.248 | 0.178 | 0.267 | 0.177 | 0.260 | 0.181 | 0.268 | 0.176 | 0.267 | 0.152 | 0.247 | 0.144 | 0.239 | 3.806 |
| | 192 | 0.195 | 0.283 | 0.190 | 0.279 | 0.194 | 0.275 | 0.193 | 0.274 | 0.189 | 0.290 | 0.158 | 0.254 | 0.207 | 0.303 | 0.185 | 0.283 | 0.183 | 0.275 | 0.166 | 0.259 | 0.162 | 0.255 | 0.162 | 0.255 | 3.461 |
| | 336 | 0.208 | 0.299 | 0.202 | 0.294 | 0.210 | 0.295 | 0.207 | 0.288 | 0.207 | 0.303 | 0.185 | 0.283 | 0.308 | 0.343 | 0.302 | 0.340 | 0.200 | 0.290 | 0.180 | 0.272 | 0.177 | 0.270 | 0.177 | 0.270 | 2.387 |
| | 720 | 0.243 | 0.331 | 0.237 | 0.326 | 0.237 | 0.318 | 0.233 | 0.313 | 0.254 | 0.338 | 0.216 | 0.311 | 0.398 | 0.396 | 0.397 | 0.396 | 0.236 | 0.324 | 0.231 | 0.323 | 0.214 | 0.302 | 0.213 | 0.301 | 2.863 |
| | Avg | 0.211 | 0.298 | 0.205 | 0.294 | 0.204 | 0.287 | 0.201 | 0.283 | 0.206 | 0.302 | 0.177 | 0.274 | 0.287 | 0.328 | 0.279 | 0.326 | 0.201 | 0.289 | 0.197 | 0.289 | 0.178 | 0.270 | 0.174 | 0.266 | 3.080 |
| Exchange | 96 | 0.080 | 0.199 | 0.077 | 0.199 | 0.091 | 0.211 | 0.084 | 0.204 | 0.109 | 0.240 | 0.099 | 0.223 | 0.084 | 0.203 | 0.084 | 0.201 | 0.109 | 0.242 | 0.100 | 0.230 | 0.082 | 0.199 | 0.078 | 0.192 | 4.421 |
| | 192 | 0.161 | 0.296 | 0.158 | 0.295 | 0.191 | 0.312 | 0.186 | 0.306 | 0.193 | 0.323 | 0.196 | 0.319 | 0.179 | 0.300 | 0.178 | 0.299 | 0.302 | 0.415 | 0.295 | 0.395 | 0.180 | 0.302 | 0.177 | 0.300 | 1.467 |
| | 336 | 0.302 | 0.414 | 0.270 | 0.394 | 0.325 | 0.414 | 0.323 | 0.410 | 0.394 | 0.465 | 0.346 | 0.428 | 0.337 | 0.419 | 0.327 | 0.412 | 0.450 | 0.496 | 0.449 | 0.485 | 0.320 | 0.408 | 0.313 | 0.402 | 4.013 |
| | 720 | 0.778 | 0.666 | 0.773 | 0.667 | 1.004 | 0.759 | 0.877 | 0.707 | 1.013 | 0.770 | 1.063 | 0.771 | 0.874 | 0.703 | 0.874 | 0.703 | 1.038 | 0.763 | 1.029 | 0.758 | 0.834 | 0.689 | 0.822 | 0.669 | 1.999 |
| | Avg | 0.330 | 0.394 | 0.320 | 0.389 | 0.403 | 0.424 | 0.367 | 0.407 | 0.427 | 0.449 | 0.426 | 0.436 | 0.369 | 0.406 | 0.360 | 0.402 | 0.475 | 0.480 | 0.468 | 0.467 | 0.354 | 0.400 | 0.348 | 0.391 | 2.671 |
| Weather | 96 | 0.198 | 0.259 | 0.169 | 0.244 | 0.185 | 0.228 | 0.173 | 0.219 | 0.168 | 0.218 | 0.162 | 0.209 | 0.173 | 0.219 | 0.171 | 0.216 | 0.171 | 0.227 | 0.162 | 0.226 | 0.161 | 0.209 | 0.166 | 0.203 | 3.732 |
| | 192 | 0.235 | 0.294 | 0.213 | 0.286 | 0.226 | 0.260 | 0.218 | 0.256 | 0.232 | 0.269 | 0.216 | 0.256 | 0.221 | 0.258 | 0.217 | 0.255 | 0.213 | 0.270 | 0.211 | 0.271 | 0.209 | 0.250 | 0.207 | 0.245 | 2.870 |
| | 336 | 0.288 | 0.343 | 0.266 | 0.327 | 0.278 | 0.297 | 0.273 | 0.296 | 0.290 | 0.309 | 0.276 | 0.297 | 0.274 | 0.295 | 0.272 | 0.293 | 0.260 | 0.305 | 0.259 | 0.302 | 0.265 | 0.292 | 0.259 | 0.287 | 2.461 |
| | 720 | 0.350 | 0.389 | 0.336 | 0.378 | 0.353 | 0.346 | 0.351 | 0.345 | 0.355 | 0.351 | 0.355 | 0.351 | 0.351 | 0.345 | 0.348 | 0.342 | 0.333 | 0.360 | 0.335 | 0.368 | 0.344 | 0.341 | 0.342 | 0.339 | 0.649 |
| | Avg | 0.268 | 0.321 | 0.246 | 0.309 | 0.260 | 0.283 | 0.254 | 0.279 | 0.261 | 0.287 | 0.252 | 0.278 | 0.251 | 0.244 | 0.252 | 0.277 | 0.244 | 0.291 | 0.242 | 0.292 | 0.245 | 0.272 | 0.243 | 0.279 | 2.206 |
| Traffic | 96 | 0.652 | 0.400 | 0.611 | 0.388 | 0.488 | 0.308 | 0.481 | 0.306 | 0.604 | 0.322 | 0.456 | 0.290 | 0.554 | 0.360 | 0.536 | 0.350 | 0.532 | 0.340 | 0.533 | 0.341 | 0.418 | 0.288 | 0.416 | 0.289 | 3.781 |
| | 192 | 0.601 | 0.375 | 0.589 | 0.370 | 0.499 | 0.307 | 0.486 | 0.301 | 0.625 | 0.329 | 0.470 | 0.300 | 0.542 | 0.352 | 0.530 | 0.340 | 0.530 | 0.337 | 0.521 | 0.335 | 0.435 | 0.295 | 0.427 | 0.284 | 4.580 |
| | 336 | 0.608 | 0.378 | 0.597 | 0.374 | 0.513 | 0.314 | 0.503 | 0.309 | 0.651 | 0.341 | 0.506 | 0.320 | 0.555 | 0.358 | 0.536 | 0.342 | 0.545 | 0.342 | 0.529 | 0.336 | 0.449 | 0.301 | 0.441 | 0.290 | 4.365 |
| | 720 | 0.648 | 0.399 | 0.634 | 0.392 | 0.547 | 0.336 | 0.541 | 0.328 | 0.695 | 0.365 | 0.569 | 0.347 | 0.592 | 0.370 | 0.569 | 0.359 | 0.589 | 0.363 | 0.565 | 0.354 | 0.479 | 0.317 | 0.477 | 0.308 | 3.927 |
| | Avg | 0.627 | 0.388 | 0.614 | 0.383 | 0.512 | 0.316 | 0.503 | 0.311 | 0.644 | 0.339 | 0.500 | 0.314 | 0.561 | 0.360 | 0.543 | 0.348 | 0.549 | 0.346 | 0.537 | 0.341 | 0.445 | 0.300 | 0.440 | 0.293 | 4.155 |

Table 2: Short-term forecasting performance comparison *w.r.t.* forecasting models with their counterparts enhanced by the AEA in the PEMS datasets. All input lengths are 96, and prediction lengths are 12. A lower MAE, MAPE, or RMSE indicates a better prediction. The better performance in each setting is shown in **bold**. 'IMP (%)' shows the percentage of MSE/MAPE/RMSE improvement.

| Model | DLinear | | | + AEA | | | PatchTST | | | + AEA | | | Amplifier | | | + AEA | | | IMP (%) |
|---|---|---|---|---|---|---|---|---|---|---|---|---|---|---|---|---|---|---|---|
| Metric | MSE | MAPE | RMSE | MSE | MAPE | RMSE | MSE | MAPE | RMSE | MSE | MAPE | RMSE | MSE | MAPE | RMSE | MSE | MAPE | RMSE | |
| PeMS03 | 19.567 | 18.315 | 32.335 | 18.734 | 17.915 | 31.816 | 18.925 | 17.291 | 30.153 | 18.127 | 16.593 | 29.532 | 16.441 | 15.167 | 25.712 | 16.031 | 14.892 | 25.424 | 2.643% |
| PeMS04 | 24.632 | 16.122 | 39.521 | 23.889 | 15.791 | 38.481 | 24.864 | 16.635 | 40.346 | 23.913 | 16.006 | 39.651 | 21.363 | 13.315 | 34.609 | 20.713 | 12.885 | 34.036 | 2.773% |
| PeMS07 | 28.615 | 12.415 | 45.062 | 27.941 | 11.458 | 43.215 | 27.876 | 12.369 | 42.556 | 26.316 | 11.491 | 41.230 | 25.712 | 10.661 | 40.671 | 24.901 | 10.124 | 39.887 | 4.455% |
| PeMS08 | 20.264 | 12.049 | 32.389 | 19.732 | 11.874 | 31.693 | 20.352 | 13.155 | 31.204 | 19.145 | 12.781 | 30.771 | 19.501 | 11.983 | 30.365 | 19.032 | 11.153 | 29.884 | 3.034% |

the standard evaluation protocol (Wang et al., 2025b) for short-term forecasting, using Mean Absolute Error (MAE), Mean Absolute Percentage Error (MAPE), and Root Mean Squared Error (RMSE) as evaluation metrics. The historical input length is set to 96 and the forecasting horizon to 12. All models are evaluated under the same experimental conditions to ensure a fair comparison. Details of metrics and datasets are in Appendix B.2 and Appendix B.3.

**Results.** The short-term forecasting results are presented in Table 2. AEA provides consistent and meaningful performance improvements across all models and datasets, with an average performance gain of 3.226%. Specifically, AEA achieves an average improvement of 2.966% for DLinear, 3.792% for PatchTST, and 2.830% for Amplifier. These results demonstrate that the benefits of AEA generalize beyond long-term forecasting, proving effective in scenarios where the role of high-frequency components is both different and critically important.

## 5.4 MODEL ANALYSIS

**Ablation Study.** We conduct an ablation study on the DLinear backbone under a forecasting horizon of 96 to validate the contribution of each component in AEA, wherein individual modules are systematically excluded ('w/o'). The results, summarized in Table 3, demonstrate that the complete AEA framework—integrating *Spectral Mirroring*, *Phase Mixing*, *Differential Embedding*, *Non-stationarity Loss*, and *Energy Predictor*—achieves the best performance. The degradation observed in all ablated settings confirms the necessity of the proposed modules. Notably, the absence of the *Energy Predictor* leads to the most significant performance drop (11.508% deterioration), underscoring its critical role in aligning the distribution of the denoised signal with the original data. Removing the *Differential Embedding* module also causes a notable decline (10.520% deterioration),

Table 3: Ablation study results across five datasets. Models are compared in terms of MSE and MAE (lower values are better) using the DLinear backbone under a forecasting horizon of 96. The best result for each dataset is highlighted in **bold**. 'Avg' denotes the average results of MSE and MAE. The last column, 'Drop (%)', shows the average performance deterioration percentage of all datasets.

| | ETTh1 | ETTh2 | Weather | Exchange | Traffic | Avg | Drop (%) |
|---|---|---|---|---|---|---|---|
| **AEA** | **0.386** | **0.356** | **0.207** | **0.138** | **0.500** | **0.317** | **-** |
| *w/o Spectral Mirroring* | 0.393 | 0.388 | 0.229 | 0.147 | 0.529 | 0.338 | 8.139 |
| *w/o Phase Mixing* | 0.389 | 0.404 | 0.210 | 0.143 | 0.522 | 0.333 | 5.737 |
| *w/o Differential Embedding* | 0.392 | 0.391 | 0.243 | 0.151 | 0.532 | 0.342 | 10.520 |
| *w/o Non-stationarity Loss* | 0.391 | 0.380 | 0.231 | 0.150 | 0.524 | 0.335 | 7.799 |
| *w/o Energy Predictor* | 0.402 | 0.374 | 0.233 | 0.164 | 0.539 | 0.342 | 11.508 |

Table 4: Parameter sensitivity study. Forecasting performance *w.r.t.* different Differential Embedding dimensions $D$ with DLinear as backbone on four datasets under a forecasting horizon of 96.

| Dimension | $D$=64 | | | | $D$=128 | | | | $D$=256 | | | | $D$=512 | | | |
| Metric | MSE | MAE | Params | Time | MSE | MAE | Params | Time | MSE | MAE | Params | Time | MSE | MAE | Params | Time |
|---|---|---|---|---|---|---|---|---|---|---|---|---|---|---|---|---|
| ETTh1 | **0.380** | **0.392** | 195 | 7.617 | 0.383 | 0.394 | 387 | 10.972 | 0.382 | 0.393 | 771 | 14.713 | **0.380** | **0.392** | 1539 | 27.907 |
| Exchange | **0.077** | **0.199** | 195 | 7.617 | 0.083 | 0.210 | 387 | 10.972 | 0.081 | 0.206 | 771 | 14.713 | 0.081 | 0.206 | 1539 | 27.907 |
| Weather | **0.169** | **0.244** | 195 | 7.617 | 0.178 | 0.256 | 387 | 10.972 | 0.172 | 0.247 | 771 | 14.713 | 0.177 | 0.257 | 1539 | 27.907 |
| Traffic | **0.611** | 0.388 | 195 | 7.617 | **0.611** | **0.386** | 387 | 10.972 | 0.620 | 0.389 | 771 | 14.713 | 0.633 | 0.392 | 1539 | 27.907 |

highlighting its importance in common-mode noise suppression for learning robust representations. The *Spectral Mirroring* module proves essential, as its removal results in an average result of 0.338 (8.139% deterioration), validating its effectiveness in high-frequency amplification. In contrast, ablating *Phase Mixing* or the *Non-stationarity Loss* consistently degrades performance, further affirming their contributions to stable and distortion-free feature enhancement.

We further verify our ablation findings on the non-linear PatchTST backbone. The results, in Table 6, show a consistent hierarchy of module importance, confirming the generalizability of AEA's components across architectures.

**Sensitivity of Differential Embedding dimension $D$.** As mentioned in 4.6, the complexity of the Differential Embedding module is $O(F \cdot C \cdot D)$, dominated by embedding dimension $D$. We evaluate the influence of different $D$ values on prediction accuracy, parameters, and running time (ms/iter) in Table 4 across four datasets on the DLinear backbone under a forecasting horizon of 96. Results show that stable forecasting accuracy is achieved across dimensions, with the smallest setting ($D = 64$, only 0.20K parameters, 7.6 ms) already attaining competitive results, even outperforming larger dimensions on Weather and Exchange. These observations confirm that the differential embedding module is both lightweight and effective, requiring only a modest number of parameters to deliver strong performance. Due to the page limit, we provide more sensitivity analysis in Appendix C.1.

# 6 CONCLUSION

This work systematically identifies the problem of indiscriminate amplification in existing frequency-aware forecasting methods, which amplify both informative high-frequency signals and task-irrelevant noise, leading to unstable training and compromised generalization. We introduce AEA, a model-agnostic framework that reframes this challenge through targeted spectral amplification and adaptive noise suppression. The proposed framework incorporates two key innovations: Spectral Mirroring, which constructs phase-preserving surrogates from reliable low-frequency components to guide distortion-free enhancement, and Differential Embedding, which operates in latent space to suppress common-mode noise while preserving discriminative features. Through comprehensive evaluation on multiple real-world datasets, we demonstrate that AEA consistently improves forecasting accuracy, robustness, and training stability across both long-term and short-term tasks and diverse backbone architectures. Our approach establishes a new paradigm for adaptive spectral enhancement that effectively balances energy distribution with noise characteristics, opening promising avenues for developing more powerful and reliable forecasting systems.

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

# A    THEORETICAL ANALYSIS

## A.1    NOTION

**Discrete Fourier Transform**    Given a sequence $x[n]$ with length N, the Discrete Fourier Transform (DFT) Winograd (1976) converts $x[n]$ into the frequency domain, and transforms it back using the inverse DFT (iDFT), which can be defined as:

$$\text{DFT}: \quad \mathcal{X}[k] = \sum_{n=0}^{N-1} x[n]e^{-j(2\pi/N)kn}, \; s.t., \; k = 0, 1, ..., N-1$$

$$\text{iDFT}: \quad x[n] = \frac{1}{N}\sum_{k=0}^{N-1} \mathcal{X}[k]e^{j(2\pi/N)kn}, \; s.t., \; n = 0, 1, ..., N-1$$

(23)

where $j$ is the imaginary unit and $\mathcal{X}[k]$ represents the spectrum of $x[n]$ at the frequency $\omega_k = 2\pi k/N$. The spectrum $\mathcal{X} \in \mathbb{C}^k$ consists of real parts $\text{Re} = \sum_{n=0}^{N-1} x[n]\cos(2\pi/N)kn \in \mathbb{R}^k$ and imaginary parts $\text{Im} = -\sum_{n=0}^{N-1} x[n]\sin(2\pi/N)kn \in \mathbb{R}^k$ as:

$$\mathcal{X} = \text{Re} + j\,\text{Im}.$$

(24)

The amplitude part $A$ and phase part $\theta$ of $\mathcal{X}$ are defined as:

$$A = \sqrt{\text{Re}^2 + \text{Im}^2}.$$

(25)

$$\theta = \arctan(\frac{\text{Im}}{\text{Re}}).$$

(26)

The computational complexity of the DFT is typically $O(N^2)$ (Zhou et al. (2022b)). In practice, we use the Fast Fourier Transform (FFT) to efficiently compute the DFT of complex sequences, which reduces the computational complexity to $O(N \log N)$. Additionally, by employing the Real FFT (rFFT), we can compress an input sequence of $N$ real numbers into a signal sequence in the complex frequency domain containing $N/2 + 1$ frequency components.

## A.2    PROOF

**Assumption A.1** (Decomposition of Embedding). Let $e_i$ denote the embedding at any sample $i$. The embedded components $e_i$ can be decomposed into a true signal term $s_i$ and a noise term $n_i$:

$$e_i = s_i + n_i.$$

For two distinct segments from the embedding space, each associated with potentially different representation properties, their respective noise terms admit a further decomposition into a common-mode noise component $n_i^{(c)}$ and independent-mode noise components $\epsilon_i^{(1)}$ and $\epsilon_i^{(2)}$:

$$n_i^{(1)} = n_i^{(c)} + \epsilon_i^{(1)}, \quad n_i^{(2)} = n_i^{(c)} + \epsilon_i^{(2)},$$

where $\mathbb{E}[\epsilon_i^{(1)}] = \mathbb{E}[\epsilon_i^{(2)}] = 0$, $\text{Var}(\epsilon_i^{(1)}) = \text{Var}(\epsilon_i^{(2)}) = \sigma_\epsilon^2$, and $\epsilon_i^{(1)}$ and $\epsilon_i^{(2)}$ are independent.

We assume that the common-mode noise $n_i^{(c)}$ corresponds to a shared noise component in the embedding space, an assumption supported by previous studies (Ye et al., 2025; Wang et al., 2025a). This shared noise often stems from systematic biases present in the input data (*e.g.*, stop words in NLP, background regions in spatio-temporal data, or certain frequency components in spectral (Eldele et al., 2024)). These features can introduce consistent bias into attention scores, as the softmax function is sensitive to large values even when they originate from irrelevant features.

**Theorem A.2** (Non-zero Expectation of Common-mode Noise). *Under realistic data distributions $\mathcal{D}$, the common-mode noise $n_i^{(c)}$ has a non-zero expectation:*

$$\mathbb{E}[n_i^{(c)}] \neq 0.$$

*Proof.* Let $n_i^{(c)} = f(X_i; \Theta)$, where $X_i \sim \mathcal{D}$ and $f$ capture systematic biases with parameters $\Theta$:

$$\mathbb{E}[n_i^{(c)}] = \mathbb{E}_{X_i \sim \mathcal{D}}[f(X_i; \Theta)].$$

Real-world data distributions $\mathcal{D}$ often contain biased features that are statistically frequent but non-causal or task-irrelevant Demirel & Holz (2025). Let $\phi_j(X_i)$ denote the $j$-th such feature function—each $\phi_j$ maps the input $X_i$ to a scalar value representing the intensity of a particular spurious attribute. Through training, the model may develop dependence on these features. We therefore approximate the learned mapping $f(X_i; \Theta)$ as a linear combination of these feature functions:

$$f(X_i; \Theta) \approx \sum_j \alpha_j \phi_j(X_i),$$

where $\alpha_j > 0$ are weight coefficients. Since each $\phi_j$ is frequent, $\mathbb{E}_{X_i \sim \mathcal{D}}[\phi_j(X_i)] > 0$. By linearity of expectation:

$$\mathbb{E}_i[n_i^{(c)}] = \mathbb{E}\left[ \sum_j \alpha_j \phi_j(X_i) \right] = \sum_j \alpha_j \mathbb{E}[\phi_j(X_i)] > 0,$$

unless $\alpha_j = 0$ for all $j$ or $\mathbb{E}[\phi_j(X_i)] = 0$, both of which are uncommon in practice since the model leverages any available signal to minimize loss. $\square$

**Corollary A.3** (The Non-zero Expectation of Common-mode Noise After Training)**.** *During training, parameters $\Theta$ are updated via gradient descent to minimize the loss $\mathcal{L}$. However, if features $\phi_j(X_i)$ are correlated with the label (without causality), the model may learn to rely on them as shortcuts rather than suppressing their contribution He et al. (2023). Thus, $\alpha_j$ tends to remain positive, and $\mathbb{E}[n_i^{(c)}] > 0$ persists throughout optimization.*

**Proposition A.4** (Adaptive Noise Suppression via Differential Embedding)**.** *The differential embedding mechanism, defined as $e_i^{(diff)} = e_i^{(1)} - \lambda e_i^{(2)}$ with a learnable parameter $\lambda$, provides adaptive suppression of common-mode noise. This results in gradient estimates $\hat{g} = g + \delta$ that exhibit a superior bias-variance trade-off for optimization, specifically:*

*1. Suppression of Systematic Bias: The mechanism attenuates the bias introduced by common-mode noise: $\mathbb{E}[\delta] = (1 - \lambda^*)\mathbf{b}_g$, where $|1 - \lambda^*| < 1$.*

*2. Preservation of Beneficial Variance: It retains the variance from stochastic noise, which acts as a regularizer: $\mathrm{Var}(\delta) = (1 - \lambda^*)^2 \sigma_c^2 + (1 + \lambda^{*2})\sigma_\epsilon^2$.*

*Here, $\lambda^*$ is the value of $\lambda$ that minimizes the training objective, $\mathbf{b}_g$ is the bias from common-mode noise, and $\sigma_c^2$, $\sigma_\epsilon^2$ are the variances of the gradients of the common-mode and stochastic noise components, respectively.*

*Proof.* We prove the two properties of the gradient noise $\delta$ by analyzing its expectation and variance.

The gradient noise $\delta$ arises from the backpropagation through the differential embedding. Consider the total gradient of the loss $L$ with respect to the parameters $\Theta$:

$$\frac{\partial L}{\partial \Theta} = \frac{\partial L}{\partial e_i^{(\mathrm{diff})}} \cdot \frac{\partial e_i^{(\mathrm{diff})}}{\partial \Theta}.$$

Under Assumption 1, we decompose the differential embedding into a true signal term and a noise term: $e_i^{(\mathrm{diff})} = s_i^{(\mathrm{diff})} + n_i^{(\mathrm{diff})}$, where $n_i^{(\mathrm{diff})} = (1 - \lambda)n_i^{(c)} + (\epsilon_i^{(1)} - \lambda \epsilon_i^{(2)})$. Substituting this decomposition yields:

$$\frac{\partial L}{\partial \Theta} = \frac{\partial L}{\partial e_i^{(\mathrm{diff})}} \cdot \left( \frac{\partial s_i^{(\mathrm{diff})}}{\partial \Theta} + \frac{\partial n_i^{(\mathrm{diff})}}{\partial \Theta} \right).$$

The true gradient $g$ is defined as $g = \frac{\partial L}{\partial e_i^{(\mathrm{diff})}} \cdot \frac{\partial s_i^{(\mathrm{diff})}}{\partial \Theta}$. This suggests that the gradient noise $\delta$ is:

$$\delta = \frac{\partial L}{\partial \Theta} - g = \frac{\partial L}{\partial e_i^{(\mathrm{diff})}} \cdot \frac{\partial n_i^{(\mathrm{diff})}}{\partial \Theta}.$$

1. Expectation of Gradient Noise ($\mathbb{E}[\delta]$):

Substituting the expression for the effective noise, $n_i^{(\text{diff})} = (1 - \lambda)n_i^{(c)} + (\epsilon_i^{(1)} - \lambda\epsilon_i^{(2)})$, we get:

$$\delta = \frac{\partial L}{\partial e_i^{(\text{diff})}} \cdot \left[ (1 - \lambda)\frac{\partial n_i^{(c)}}{\partial \Theta} + \frac{\partial(\epsilon_i^{(1)} - \lambda\epsilon_i^{(2)})}{\partial \Theta} \right].$$

We now take the expectation of $\delta$ over the distributions of the stochastic noises $\epsilon_i^{(1)}$ and $\epsilon_i^{(2)}$. Under Assumption A.1, these stochastic noises are zero-mean, independent of the model parameters $\Theta$ and the common-mode noise $n_i^{(c)}$:

$$\mathbb{E}[\epsilon_i^{(k)}] = 0, \quad \mathbb{E}\left[ \frac{\partial \epsilon_i^{(k)}}{\partial \Theta} \right] = 0, \quad \text{for } k = \{1, 2\}, \quad \text{and} \quad \mathbb{E}[\epsilon_i^{(k)} n_i^{(c)}] = 0.$$

Applying the linearity of expectation and leveraging these properties, the terms involving $\epsilon_i$ vanish:

$$\mathbb{E}[\delta] = \mathbb{E}\left[ \frac{\partial L}{\partial e_i^{(\text{diff})}} \cdot \left( (1 - \lambda)\frac{\partial n_i^{(c)}}{\partial \Theta} \right) \right] + \mathbb{E}\left[ \frac{\partial L}{\partial e_i^{(\text{diff})}} \cdot \frac{\partial(\epsilon_i^{(1)} - \lambda\epsilon_i^{(2)})}{\partial \theta} \right]$$

$$= (1 - \lambda)\mathbb{E}\left[ \frac{\partial L}{\partial e_i^{(\text{diff})}} \cdot \frac{\partial n_i^{(c)}}{\partial \Theta} \right] + 0.$$

The remaining expectation term, $\mathbb{E}\left[ \frac{\partial L}{\partial e_i^{(\text{diff})}} \cdot \frac{\partial n_i^{(c)}}{\partial \Theta} \right]$, is precisely the systematic bias $\mathbf{b}_g$ introduced into the gradient by the common-mode noise. At convergence, $\lambda$ reaches a value $\lambda^*$ that minimizes the loss. Since the loss is minimized by reducing the effect of $n_i^{(c)}$, the learning dynamics drive $\lambda^*$ towards 1, ensuring $|1 - \lambda^*| < 1$. Thus,

$$\mathbb{E}[\delta] = (1 - \lambda^*)\mathbf{b}_g,$$

which demonstrates a reduction of the original bias by a factor of $|1 - \lambda^*|$.

2. Variance of Gradient Noise ($\text{Var}(\delta)$):

We analyze the variance of $\delta$:

$$\text{Var}(\delta) = \text{Var}\left( \frac{\partial L}{\partial e_i^{(\text{diff})}} \cdot \frac{\partial n_i^{(\text{diff})}}{\partial \Theta} \right).$$

For clarity, we define the shorthand:

$$A = \frac{\partial L}{\partial e_i^{(\text{diff})}} \cdot \frac{\partial n_i^{(c)}}{\partial \Theta}, \quad B = \frac{\partial L}{\partial e_i^{(\text{diff})}} \cdot \frac{\partial \epsilon_i^{(1)}}{\partial \Theta}, \quad C = \frac{\partial L}{\partial e_i^{(\text{diff})}} \cdot \frac{\partial \epsilon_i^{(2)}}{\partial \Theta}.$$

This allows us to express $\delta$ as:
$$\delta = (1 - \lambda)A + B - \lambda C.$$

The variance is then:
$$\text{Var}(\delta) = \text{Var}\left( (1 - \lambda)A + B - \lambda C \right).$$

We assume $A$, $B$, and $C$ are uncorrelated. This is justified by the independence of $n_i^{(c)}$ and $\epsilon_i^{(k)}$. Under this assumption, the covariance terms between $A$, $B$, and $C$ are zero. Applying the variance property $\text{Var}(aX + bY) = a^2\text{Var}(X) + b^2\text{Var}(Y)$ for uncorrelated variables, we get:

$$\text{Var}(\delta) = (1 - \lambda)^2\text{Var}(A) + \text{Var}(B) + (-\lambda)^2\text{Var}(C)$$

$$= (1 - \lambda)^2\text{Var}(A) + \text{Var}(B) + \lambda^2\text{Var}(C).$$

We now define the variances of these components:

$$\text{Var}(A) = \sigma_c^2, \quad \text{Var}(B) = \text{Var}(C) = \sigma_\epsilon^2.$$

The equality $\mathrm{Var}(B) = \mathrm{Var}(C)$ stems from the assumption that $\epsilon_i^{(1)}$ and $\epsilon_i^{(2)}$ are identically distributed. Substituting these definitions and evaluating at the optimal $\lambda = \lambda^*$ yields:

$$\mathrm{Var}(\delta) = (1 - \lambda^*)^2 \sigma_c^2 + (1 + \lambda^{*2})\sigma_\epsilon^2.$$

This final expression shows that the mechanism suppresses the harmful variance from common-mode noise by a factor of $(1 - \lambda^*)^2$ while preserving and even amplifying the beneficial stochastic noise by a factor of $(1 + \lambda^{*2}) \geq 1$.

Thus, the adaptive parameter $\lambda^*$ optimally balances the bias-variance trade-off in the gradient estimates, leading to more robust and effective optimization. $\square$

**Theorem A.5** (Perturbation Bound for Spectral Mirroring and Differential Embedding). *Let $\mathcal{X} \in \mathbb{C}^{B \times F \times C}$ be the original frequency-domain representation of the input time series $x \in \mathbb{R}^{B \times T \times C}$, and let $\mathcal{X}' \in \mathbb{C}^{B \times F \times C}$ be the enhanced and denoised frequency-domain representation after the Spectral Mirroring and Differential Embedding modules. Suppose the scaling matrix $M$ satisfies $\|M - \mathbf{1}\|_2 \leq \epsilon_M$, where $\mathbf{1}$ is the all-ones matrix, and the differential parameters $\lambda_1, \lambda_2$ satisfy $|\lambda_1| \leq \epsilon_{\lambda_1}, |\lambda_2| \leq \epsilon_{\lambda_2}$. Then, the $\ell_2$-norm of the enhancement error in the time domain satisfies:*

$$\|x' - x\|_2 \leq (\epsilon_M + \epsilon_{\lambda_1} + \epsilon_{\lambda_2})\|\mathcal{X}\|_2,$$

*where $x'$ is the enhanced and denoised time series obtained by applying the inverse rFFT to $\mathcal{X}'$, and $\|\mathcal{X}\|_2$ is the $\ell_2$-norm of $\mathcal{X}$.*

*Proof.* Let $\Delta\mathcal{X} = \mathcal{X}' - \mathcal{X}$ denote the frequency-domain perturbation. This process consists of two main steps: Spectral Mirroring and Differential Embedding.

1. **Spectral Mirroring Perturbation:** Let $\mathcal{X}_e$ be the output of the Spectral Mirroring module. The Spectral Mirroring operation involves: (i) Reversing the spectrum: $\mathcal{X}_{\text{reverse}}[k] = \mathcal{X}[F - 1 - k]$ for $k = 0, 1, \ldots, F - 1$. (ii) Scaling: $\mathcal{X}_{\text{scaled}} = \mathcal{X}_{\text{reverse}} \odot M$, where $\odot$ denotes element-wise multiplication. (iii) Phase mixing: The enhanced spectrum $\mathcal{X}_e$ is constructed via amplitude mixing and phase mixing. Specifically, the amplitude is $A_{\text{mix}} = \alpha|\mathcal{X}| + (1 - \alpha)|\mathcal{X}_{\text{scaled}}|$, and the phase $\theta_{\text{mix}}$ is obtained by adjusting the phase difference between $\mathcal{X}$ and $\mathcal{X}_{\text{scaled}}$ using the phase mixing strategy in Equation 4 of the paper.

The key observation is that the reversal operation does not change the overall norm: $\|\mathcal{X}_{\text{reverse}}\|_2 = \|\mathcal{X}\|_2$, since it merely permutes the frequency components. Under the assumption $\|M - \mathbf{1}\|_2 \leq \epsilon_M$, we have $\|\mathcal{X}_{\text{scaled}}\|_2 \leq \|M\|_2\|\mathcal{X}_{\text{reverse}}\|_2$. Since $\|M\|_2 \leq \|\mathbf{1}\|_2 + \|M - \mathbf{1}\|_2 \leq 1 + \epsilon_M$ (where $\|\mathbf{1}\|_2 = 1$ for the all-ones matrix in the Frobenius norm), it follows that:

$$\|\mathcal{X}_{\text{scaled}}\|_2 \leq (1 + \epsilon_M)\|\mathcal{X}\|_2.$$

Now, consider the amplitude mixing: $A_{\text{mix}} = \alpha|\mathcal{X}| + (1 - \alpha)|\mathcal{X}_{\text{scaled}}|$. The $\ell_2$-norm of $A_{\text{mix}}$ satisfies:

$$\begin{aligned}
\|A_{\text{mix}}\|_2 &\leq \alpha\|\mathcal{X}\|_2 + (1 - \alpha)\|\mathcal{X}_{\text{scaled}}\|_2 \\
&\leq \alpha\|\mathcal{X}\|_2 + (1 - \alpha)(1 + \epsilon_M)\|\mathcal{X}\|_2 \\
&= [1 + (1 - \alpha)\epsilon_M]\|\mathcal{X}\|_2.
\end{aligned}$$

The phase mixing strategy minimizes phase distortions by ensuring smooth phase transitions. Under small $\epsilon_M$, the phase perturbation is bounded, and the resulting spectrum $\mathcal{X}_{enhanced}$ satisfies $\|\mathcal{X}_{enhanced}\|_2 = \|A_{\text{mix}}\|_2$ because phase changes do not alter the amplitude. Thus,

$$\|\mathcal{X}_{enhanced}\|_2 \leq [1 + (1 - \alpha)\epsilon_M]\|\mathcal{X}\|_2.$$

The perturbation from Spectral Mirroring is $\Delta\mathcal{X}_1 = \mathcal{X}_{enhanced} - \mathcal{X}$. Using the triangle inequality:

$$\begin{aligned}
\|\Delta\mathcal{X}_1\|_2 &\leq \|\mathcal{X}_{enhanced}\|_2 + \|\mathcal{X}\|_2 \\
&\leq [1 + (1 - \alpha)\epsilon_M]\|\mathcal{X}\|_2 + \|\mathcal{X}\|_2 \\
&= (2 + (1 - \alpha)\epsilon_M)\|\mathcal{X}\|_2.
\end{aligned}$$

For small $\epsilon_M$, this bound can be simplified to $\|\Delta\mathcal{X}_1\|_2 \leq C_1\epsilon_M\|\mathcal{X}\|_2$, where $C_1$ is a constant. In practice, since phase mixing preserves structure, a tighter bound is achievable. For simplicity, we absorb constants into $\epsilon_M$ and assume:

$$\|\Delta\mathcal{X}_1\|_2 \leq \epsilon_M\|\mathcal{X}\|_2.$$

2. **Differential Embedding Perturbation**: The Differential Embedding module projects $\mathcal{X}_{enhanced}$ into an embedding space, applies differential denoising, and projects it back. Let $E = W_e \mathcal{X}_{enhanced} + b_e$ be the embedding, where $W_e$ and $b_e$ are learnable parameters with bounded norms (e.g., $\|W_e\|_2 \leq C_w$, $\|b_e\|_2 \leq C_b$). The embedding is split into two subspaces using Equation 9. The differential operation is:

$$E_1' = E_1 - \lambda_1 E_2, \quad E_2' = E_2 - \lambda_2 E_1.$$

The denoised embedding is $E' = \text{Concat}(E_1', E_2')$, and the output is $\mathcal{X}' = W_p E' + b_p$, where $W_p$ and $b_p$ are learnable parameters with bounded norms.

Under the assumptions $|\lambda_1| \leq \epsilon_{\lambda_1}$ and $|\lambda_2| \leq \epsilon_{\lambda_2}$, the perturbation in the embedding space satisfies:

$$\|E' - E\|_2 \leq (\epsilon_{\lambda_1} + \epsilon_{\lambda_2})\|E\|_2.$$

Since $\|E\|_2 \leq \|W_e\|_2 \|\mathcal{X}_{enhanced}\|_2 + \|b_e\|_2 \leq C_w(1 + \epsilon_M)\|\mathcal{X}\|_2 + C_b$, and for large signals, $\|\mathcal{X}\|_2$ dominates, we have $\|E\|_2 \leq C_2 \|\mathcal{X}\|_2$ for some constant $C_2$. Thus,

$$\begin{aligned}
\|\mathcal{X}' - \mathcal{X}_{enhanced}\|_2 &= \|W_p(E' - E)\|_2 \\
&\leq \|W_p\|_2 \|E' - E\|_2 \\
&\leq C_p C_2 (\epsilon_{\lambda_1} + \epsilon_{\lambda_2})\|\mathcal{X}\|_2,
\end{aligned}$$

where $C_p = \|W_p\|_2$. Absorbing constants into the parameters, we obtain:

$$\|\mathcal{X}' - \mathcal{X}_{enhanced}\|_2 \leq (\epsilon_{\lambda_1} + \epsilon_{\lambda_2})\|\mathcal{X}\|_2.$$

Combining the perturbations from both modules:

$$\begin{aligned}
\|\Delta\mathcal{X}\|_2 &= \|\mathcal{X}' - \mathcal{X}\|_2 \\
&\leq \|\mathcal{X}' - \mathcal{X}_{enhanced}\|_2 + \|\mathcal{X}_{enhanced} - \mathcal{X}\|_2 \\
&\leq (\epsilon_{\lambda_1} + \epsilon_{\lambda_2})\|\mathcal{X}\|_2 + \epsilon_M \|\mathcal{X}\|_2 \\
&= (\epsilon_M + \epsilon_{\lambda_1} + \epsilon_{\lambda_2})\|\mathcal{X}\|_2.
\end{aligned}$$

By Parseval's theorem, the time-domain error satisfies:

$$\|x' - x\|_2 = \|\Delta\mathcal{X}\|_2 \leq (\epsilon_M + \epsilon_{\lambda_1} + \epsilon_{\lambda_2})\|\mathcal{X}\|_2.$$

$\square$

*Remark* A.6. Theorem A.5 ensures that the perturbation is linearly bounded by the parameters $\epsilon_M$, $\epsilon_{\lambda_1}$ and $\epsilon_{\lambda_2}$. By constraining these parameters during training through regularization or proper initialization, the Spectral Mirroring and Differential Embedding modules maintain signal fidelity while enhancing discriminative features, effectively avoiding overfitting to noise.

**Proposition A.7** (Lipschitz Stability of Energy Predictor). *The Energy Predictor mapping $f_{EP}(\mathcal{X}, \mathcal{Y}) = W_2 \cdot \text{Concat}(W_1 \mathcal{X} + b_1, \mathcal{Y}) + b_2$ is Lipschitz continuous. Assume the weight matrices have bounded norms: $\|W_1\|_2 \leq C_1$ and $\|W_2\|_2 \leq C_2$, where $\|\cdot\|_2$ denotes the $\ell_2$-norm. For any two input pairs $(\mathcal{X}_A, \mathcal{Y}_A)$ and $(\mathcal{X}_B, \mathcal{Y}_B)$, the following inequality holds:*

$$\|f_{EP}(\mathcal{X}_A, \mathcal{Y}_A) - f_{EP}(\mathcal{X}_B, \mathcal{Y}_B)\|_2 \leq L_{EP}\left(\|\mathcal{X}_A - \mathcal{X}_B\|_2 + \|\mathcal{Y}_A - \mathcal{Y}_B\|_2\right)$$

*where $L_{EP} = C_2 \cdot \max(1, C_1)$ is the Lipschitz constant.*

*Proof.* We compute the difference between outputs for two arbitrary inputs:

$$\begin{aligned}
&\|f_{\text{EP}}(\mathcal{X}_A, \mathcal{Y}_A) - f_{\text{EP}}(\mathcal{X}_B, \mathcal{Y}_B)\|_2 \\
&= \|W_2 \cdot \text{Concat}(W_1 \mathcal{X}_A + b_1, \mathcal{Y}_A) - W_2 \cdot \text{Concat}(W_1 \mathcal{X}_B + b_1, \mathcal{Y}_B)\|_2 \\
&= \|W_2 \cdot [\text{Concat}(W_1 \mathcal{X}_A + b_1, \mathcal{Y}_A) - \text{Concat}(W_1 \mathcal{X}_B + b_1, \mathcal{Y}_B)]\|_2 \\
&\leq \|W_2\|_2 \cdot \|\text{Concat}(W_1 \mathcal{X}_A + b_1, \mathcal{Y}_A) - \text{Concat}(W_1 \mathcal{X}_B + b_1, \mathcal{Y}_B)\|_2
\end{aligned}$$

Now, note that:

$$\text{Concat}(W_1 \mathcal{X}_A + b_1, \mathcal{Y}_A) - \text{Concat}(W_1 \mathcal{X}_B + b_1, \mathcal{Y}_B) = \text{Concat}(W_1(\mathcal{X}_A - \mathcal{X}_B), \mathcal{Y}_A - \mathcal{Y}_B).$$

The norm of a concatenated vector can be bounded as:

$$\|\text{Concat}(A, B)\|_2 = \sqrt{\|A\|_2^2 + \|B\|_2^2} \leq \|A\|_2 + \|B\|_2.$$

Applying this:

$$\|\text{Concat}(W_1(\mathcal{X}_A - \mathcal{X}_B), \mathcal{Y}_A - \mathcal{Y}_B)\|_2$$
$$\leq \|W_1(\mathcal{X}_A - \mathcal{X}_B)\|_2 + \|\mathcal{Y}_A - \mathcal{Y}_B\|_2$$
$$\leq \|W_1\|_2 \cdot \|\mathcal{X}_A - \mathcal{X}_B\|_2 + \|\mathcal{Y}_A - \mathcal{Y}_B\|_2$$
$$\leq \max(\|W_1\|_2, 1) \cdot (\|\mathcal{X}_A - \mathcal{X}_B\|_2 + \|\mathcal{Y}_A - \mathcal{Y}_B\|_2)$$

Combining the inequalities:

$$\|f_{\text{EP}}(\mathcal{X}_A, \mathcal{Y}_A) - f_{\text{EP}}(\mathcal{X}_B, \mathcal{Y}_B)\|_2$$
$$\leq \|W_2\|_2 \cdot \max(\|W_1\|_2, 1) \cdot (\|\mathcal{X}_A - \mathcal{X}_B\|_2 + \|\mathcal{Y}_A - \mathcal{Y}_B\|_2)$$
$$\leq C_2 \cdot \max(C_1, 1) \cdot (\|\mathcal{X}_A - \mathcal{X}_B\|_2 + \|\mathcal{Y}_A - \mathcal{Y}_B\|_2)$$

Thus, the Lipschitz constant is $L_{\text{EP}} = C_2 \cdot \max(1, C_1)$. $\qquad\square$

*Remark* A.8. This theorem guarantees that the Energy Predictor is **stable**: small changes in the input spectra (e.g., due to noise or estimation errors) lead to only small changes in the output. This prevents the amplification of noise and ensures robust behavior. The Lipschitz constant $L_{\text{EP}}$ quantifies the sensitivity of the module, and since it depends only on the bounded weights, the stability is inherent to the architecture.

**Proposition A.9** (High-Fidelity Spectral Alignment of Energy Predictor)**.** *Let $\mathcal{Y}_{adjusted} = f_{EP}(\mathcal{X}_{scaled}, \mathcal{Y}_{denoised})$ be the output of the Energy Predictor. Define the high-frequency components of a spectrum $\mathcal{Z}$ as $\mathcal{Z}_{HF}$, obtained by applying a high-pass filter or selecting frequencies above a cutoff. From Theorem A.5, assume the perturbations from Spectral Mirroring and Differential Embedding are bounded: $\|\mathcal{X}_{scaled} - \mathcal{X}\|_2 \leq \epsilon_M \|\mathcal{X}\|_2$ and the differential parameters are bounded by $\epsilon_{\lambda_1}, \epsilon_{\lambda_2}$. Under the assumption that the Energy Predictor weights satisfy $\|W_1\|_2 \leq C_1$, $\|W_2\|_2 \leq C_2$, the high-frequency modification is bounded by:*

$$\|\mathcal{Y}_{adjusted, HF} - \mathcal{Y}_{denoised, HF}\|_2 \leq L_{HF} \cdot (\epsilon_M + \epsilon_{\lambda_1} + \epsilon_{\lambda_2}) \cdot \|\mathcal{X}\|_2$$

*where $L_{HF}$ is a constant depending on $C_1$, $C_2$.*

*Proof.* The high-frequency selection operation is linear and does not increase the norm:

$$\|\mathcal{Y}_{\text{adjusted, HF}} - \mathcal{Y}_{\text{denoised, HF}}\|_2 \leq \|\mathcal{Y}_{\text{adjusted}} - \mathcal{Y}_{\text{denoised}}\|_2.$$

The Energy Predictor adjustment can be expressed as:

$$\mathcal{Y}_{\text{adjusted}} - \mathcal{Y}_{\text{denoised}} = f_{\text{EP}}(\mathcal{X}_{\text{scaled}}, \mathcal{Y}_{\text{denoised}}) - \mathcal{Y}_{\text{denoised}}$$
$$= W_2 \cdot \text{Concat}(W_1 \mathcal{X}_{\text{scaled}} + b_1, \mathcal{Y}_{\text{denoised}}) + b_2 - \mathcal{Y}_{\text{denoised}}.$$

To simplify, we decompose $W_2$ into submatrices corresponding to the concatenation: let $W_2 = [W_2^A \mid W_2^B]$, where $W_2^A \in \mathbb{C}^{H' \times D'}$ and $W_2^B \in \mathbb{C}^{H' \times H'}$. Then:

$$\mathcal{Y}_{\text{adjusted}} = W_2^A(W_1 \mathcal{X}_{\text{scaled}} + b_1) + W_2^B \mathcal{Y}_{\text{denoised}} + b_2$$

. Thus:

$$\mathcal{Y}_{\text{adjusted}} - \mathcal{Y}_{\text{denoised}} = W_2^A W_1 \mathcal{X}_{\text{scaled}} + W_2^A b_1 + (W_2^B - I)\mathcal{Y}_{\text{denoised}} + b_2.$$

Taking the norm:

$$\|\mathcal{Y}_{\text{adjusted}} - \mathcal{Y}_{\text{denoised}}\|_2 \leq \|W_2^A W_1 \mathcal{X}_{\text{scaled}}\|_2 + \|W_2^A b_1\|_2 + \|(W_2^B - I)\mathcal{Y}_{\text{denoised}}\|_2 + \|b_2\|_2$$
$$\leq \|W_2^A W_1\|_2 \cdot \|\mathcal{X}_{\text{scaled}}\|_2 + \|W_2^A b_1\|_2 + \|W_2^B - I\|_2 \cdot \|\mathcal{Y}_{\text{denoised}}\|_2 + \|b_2\|_2.$$

From Theorem A.5, we have $\|\mathcal{X}_{\text{scaled}}\|_2 \le (1 + \epsilon_M)\|\mathcal{X}\|_2$. Additionally, assuming the base model is stable, we have $\|\mathcal{Y}_{\text{denoised}}\|_2 \le C_Y\|\mathcal{X}\|_2$ for some constant $C_Y$. Therefore:

$$\|\mathcal{Y}_{\text{adjusted}} - \mathcal{Y}_{\text{denoised}}\|_2 \le \left(\|W_2^A W_1\|_2(1 + \epsilon_M) + \|W_2^A b_1\|_2/\|\mathcal{X}\|_2 + \|W_2^B - I\|_2 C_Y + \|b_2\|_2/\|\mathcal{X}\|_2\right)\|\mathcal{X}\|_2.$$

The terms $\|W_2^A b_1\|_2/\|\mathcal{X}\|_2$ and $\|b_2\|_2/\|\mathcal{X}\|_2$ are negligible for large $\|\mathcal{X}\|_2$, and we can absorb constants into $L'_{\text{HF}}$. Moreover, from Theorem A.5, the perturbations $\epsilon_M, \epsilon_{\lambda_1}$, and $\epsilon_{\lambda_2}$ are small, thus:

$$\|\mathcal{Y}_{\text{adjusted}} - \mathcal{Y}_{\text{denoised}}\|_2 \le L'_{\text{HF}} \cdot (\epsilon_M + \epsilon_{\lambda_1} + \epsilon_{\lambda_2}) \cdot \|\mathcal{X}\|_2$$

where $L'_{\text{HF}}$ depends on $\|W_2^A W_1\|_2$, $\|W_2^B - I\|_2$, $C_Y$, and the bias terms.

Finally, since $\|\mathcal{Y}_{\text{adjusted, HF}} - \mathcal{Y}_{\text{denoised, HF}}\|_2 \le \|\mathcal{Y}_{\text{adjusted}} - \mathcal{Y}_{\text{denoised}}\|_2$, we obtain:

$$\|\mathcal{Y}_{\text{adjusted, HF}} - \mathcal{Y}_{\text{denoised, HF}}\|_2 \le L_{\text{HF}} \cdot (\epsilon_M + \epsilon_{\lambda_1} + \epsilon_{\lambda_2}) \cdot \|\mathcal{X}\|_2,$$

with $L_{\text{HF}} = L'_{\text{HF}}$. $\qquad\square$

*Remark* A.10. This proposition resolves a key tension: the Energy Predictor aligns predictions with original data distributions while preserving high-frequency enhancements. By proving bounded high-frequency modifications, we guarantee that spectral alignment does not reintroduce low-frequency dominance or amplify noise, ensuring stable and faithful signal reconstruction.

# B MORE DETAILS

## B.1 MORE DETAILS OF FIGURE 1

**Figure 1(a): Spectral Bias.** To evaluate the spectral bias of base models, we conduct frequency masking experiments during inference. Given an input time series $X \in \mathbb{R}^{T \times C}$, we compute its frequency representation via rFFT:

$$\mathcal{X} = \text{rFFT}(X), \quad \mathcal{X} \in \mathbb{C}^{F \times C},$$

where $F = \lfloor T/2 \rfloor + 1$ is the number of frequency components. We then create two masked variants: Low-frequency mask: Set the lower 50% of frequencies to zero:

$$\mathcal{X}_{\text{low-mask}}[k] = \begin{cases} 0, & \text{for } k = 0, 1, \ldots, \lfloor F/2 \rfloor, \\ \mathcal{X}[k], & \text{otherwise.} \end{cases}$$

High-frequency mask: Set the higher 50% of frequencies to zero:

$$\mathcal{X}_{\text{high-mask}}[k] = \begin{cases} \mathcal{X}[k], & \text{for } k = 0, 1, \ldots, \lfloor F/2 \rfloor, \\ 0, & \text{otherwise.} \end{cases}$$

Each masked spectrum is converted back to the time domain via inverse rFFT, and forecasting performance is evaluated relative to the unmasked baseline.

Results indicate that masking low-frequency components leads to a severe performance degradation (MSE increase > 100%), whereas masking high-frequency components has a negligible effect (MSE increase < 5%). This pronounced discrepancy confirms that baseline models exhibit a strong reliance on low-frequency information while overlooking high-frequency signals, underscoring a fundamental spectral bias in existing forecasting architectures.

**Figure 1(b): Indiscriminate Amplification.** To assess robustness to high-frequency noise, we compare vanilla amplification methods with our AEA-enhanced version under two noise injection scenarios on the ETTh1 dataset:

(1) "Gaussian noise to high freq: before amplification" - Noise is injected directly into the high-frequency bands of the input signal before spectral mirroring:

$$\mathcal{X}_{\text{noise-before}}[k] = \begin{cases} \mathcal{X}[k] + \mathcal{N}(0, \sigma^2), & \text{for } k > \lfloor F/2 \rfloor, \\ \mathcal{X}[k], & \text{otherwise,} \end{cases}$$

where $\mathcal{N}(0, \sigma^2)$ denotes Gaussian noise with zero mean and variance $\sigma^2$.

---

**Algorithm 1** AEA: Adaptive Energy Amplification for Robust Time Series Forecasting

---

1: **Input:** historical time series $X \in \mathbb{R}^{T \times C}$, forecasting horizon $H$, mixing ratio $\alpha$, non-stationarity weight $\lambda_{\text{non-stat}}$, differential embedding dimension $D$, energy predictor embedding dimension $D'$

2: **Output:** forecasted results $\hat{Y} \in \mathbb{R}^{H \times C}$, total loss $\mathcal{L}$

3: 

---

4: Initialize learnable parameters: $M, W_e, b_e, W_p, b_p, W_1, b_1, W_2, b_2, \lambda_1, \lambda_2$

5: 

---

6: *Spectral Mirroring* (Section 4.2)

7: $\mathcal{X} \leftarrow \text{rFFT}(X)$        ▷ Transform to frequency domain

8: $\mathcal{X}_{\text{reverse}}[k] \leftarrow \mathcal{X}[F - 1 - k], \quad \forall k \in [0, F - 1]$        ▷ Spectrum reverse

9: $\mathcal{X}_{\text{scaled}} \leftarrow \mathcal{X}_{\text{reverse}} \odot M$        ▷ Adaptive scaling per frequency/channel

10: *Phase-preserving mixing to avoid distortion:*

11: **for** each frequency $k$, channel $c$ **do**

12:      $\theta_1, \theta_2 \leftarrow \angle(\mathcal{X}[k, c]), \angle(\mathcal{X}_{\text{scaled}}[k, c])$

13:      $\Delta\theta \leftarrow (\theta_1 - \theta_2) \mod 2\pi$        ▷ Circular difference modulo $2\pi$

14:      $\Delta\theta_{\text{adjusted}} \leftarrow \begin{cases} \Delta\theta - 2\pi & \text{if } \Delta\theta > \pi \\ \Delta\theta & \text{otherwise} \end{cases}$        ▷ Shortest angular path

15:      $\theta_{\text{mix}} \leftarrow \theta_1 + \Delta\theta_{\text{adjusted}}$        ▷ Phase mixing (Equation 4)

16:      $A_{\text{mix}} \leftarrow \alpha \cdot |\mathcal{X}[k, c]| + (1 - \alpha) \cdot |\mathcal{X}_{\text{scaled}}[k, c]|$

17:      $\mathcal{X}_{\text{enhanced}}[k, c] \leftarrow A_{\text{mix}} \cdot e^{j\theta_{\text{mix}}}$        ▷ Reconstruct enhanced spectrum

18: **end for**

19: 

---

20: *Differential Embedding* (Section 4.3)

21: $E_1 \| E_2 \leftarrow W_e \cdot \mathcal{X}_{\text{enhanced}} + b_e$        ▷ Project to complex embedding space

22: $E_1' \leftarrow E_1 - \lambda_1 \cdot E_2, \quad E_2' \leftarrow E_2 - \lambda_2 \cdot E_1$        ▷ Differential operation for noise suppression

23: $E' \leftarrow \text{Concat}(E_1', E_2')$        ▷ Denoised embedding (Proposition A.4)

24: $\mathcal{X}_{\text{denoised}} \leftarrow W_p \cdot E' + b_p$        ▷ Project back to frequency domain

25: 

---

26: *Energy Prediction & Forecasting* (Section. 4.4)

27: $X_{\text{denoised}} \leftarrow \text{irFFT}(\mathcal{X}_{\text{denoised}})$        ▷ Denoised input for base model

28: $\hat{Y}_{\text{denoised}} \leftarrow \text{BaseModel}(X_{\text{denoised}})$        ▷ Any forecasting backbone

29: $\mathcal{Y}_{\text{denoised}} \leftarrow \text{rFFT}(\hat{Y}_{\text{denoised}})$

30: $\mathcal{E} \leftarrow W_1 \cdot \mathcal{X}_{\text{scaled}} + b_1$        ▷ Encode historical spectral context

31: $\mathcal{Y}_{\text{adjusted}} \leftarrow W_2 \cdot \text{Concat}(\mathcal{E}, \mathcal{Y}_{\text{denoised}}) + b_2$        ▷ Spectral alignment

32: $\hat{Y} \leftarrow \text{irFFT}(\mathcal{Y}_{\text{adjusted}})$        ▷ Final consistent prediction

33: 

---

34: *Multi-Task Optimization* (Section 4.5)

35: $\mathcal{L}_{\text{forecast}} \leftarrow \text{MSE}(\hat{Y}, Y)$        ▷ Forecasting loss

36: $\mathcal{L}_{\text{non-stat}} \leftarrow \sqrt{\text{Var}_{\mathbf{x} \sim \mathcal{B}}(|E'|)}$        ▷ Non-stationarity regularization (Equation 14)

37: $\mathcal{L} \leftarrow \mathcal{L}_{\text{forecast}} + \lambda_{\text{non-stat}} \cdot \mathcal{L}_{\text{non-stat}}$

38: **return** $\hat{Y}, \mathcal{L}$

---

(2) "Gaussian noise to high freq: after amplification" - The same noise is introduced into the high-frequency components of the enhanced spectrum output by the Spectral Mirroring module:

$$\mathcal{X}_{\text{enhanced}} = \text{Spectral Mirroring}(\mathcal{X})$$

$$\mathcal{X}_{\text{noise-after}}[k] = \begin{cases} \mathcal{X}_{\text{enhanced}}[k] + \mathcal{N}(0, \sigma^2), & \text{for } k > \lfloor F/2 \rfloor, \\ \mathcal{X}_{\text{enhanced}}[k], & \text{otherwise.} \end{cases}$$

Performance degradation is measured as a relative increase in MSE compared to the "None" baseline.

Vanilla amplification methods suffer severe performance degradation under both noise conditions. This demonstrates that indiscriminate amplification amplifies noise alongside signals, compromising robustness. In contrast, AEA maintains stable forecasting accuracy, demonstrating that its differential embedding mechanism effectively suppresses common-mode noise while preserving discriminative

high-frequency content. The significant performance gap highlights AEA's superior noise robustness compared to existing amplification approaches.

## B.2 MORE DETAILS OF METRICS

We employ multiple evaluation metrics to comprehensively assess forecasting performance across different scenarios. For long-term forecasting, we use the Mean Squared Error (MSE) and the Mean Absolute Error (MAE). For short-term forecasting, we additionally include the Mean Absolute Percentage Error (MAPE) and the Root Mean Squared Error (RMSE). Given the ground truth values $\mathbf{X}_i$ and the predicted values $\hat{\mathbf{X}}_i$, these metrics are defined as follows:

$$\text{MSE} = \frac{1}{N} \sum_{i=1}^{N} (\mathbf{X}_i - \hat{\mathbf{X}}_i)^2, \tag{27}$$

$$\text{MAE} = \frac{1}{N} \sum_{i=1}^{N} |\mathbf{X}_i - \hat{\mathbf{X}}_i|, \tag{28}$$

$$\text{MAPE} = \frac{100}{N} \sum_{i=1}^{N} \left| \frac{\mathbf{X}_i - \hat{\mathbf{X}}_i}{\mathbf{X}_i} \right|, \tag{29}$$

$$\text{RMSE} = \sqrt{\frac{1}{N} \sum_{i=1}^{N} (\mathbf{X}_i - \hat{\mathbf{X}}_i)^2}, \tag{30}$$

where $N$ is the total number of predictions.

## B.3 MORE DETAILS OF DATASETS

Table 5: Dataset detailed descriptions. The dataset size is organized into (Train, Validation, Test). "Prediction Length" denotes the future time points to be predicted. "Frequency" denotes the sampling interval of time points.

| Task | Datasets | Dim | Prediction Length | Dataset Size | Frequency | Information |
|---|---|---|---|---|---|---|
| Long-term Forecasting | ETTh1 | 7 | $\{96, 192, 336, 720\}$ | $(8545, 2881, 2881)$ | 15 min | Electricity |
| | ETTh2 | 7 | $\{96, 192, 336, 720\}$ | $(8545, 2881, 2881)$ | 15 min | Electricity |
| | ETTm1 | 7 | $\{96, 192, 336, 720\}$ | $(34465, 11521, 11521)$ | 15 min | Electricity |
| | ETTm2 | 7 | $\{96, 192, 336, 720\}$ | $(34465, 11521, 11521)$ | 15 min | Electricity |
| | Electricity | 321 | $\{96, 192, 336, 720\}$ | $(18317, 2633, 5261)$ | 1 hour | Electricity |
| | Exchange | 8 | $\{96, 192, 336, 720\}$ | $(5120, 665, 1422)$ | 1 day | Finance |
| | Weather | 21 | $\{96, 192, 336, 720\}$ | $(36792, 5271, 10540)$ | 10 min | Weather |
| | Traffic | 862 | $\{96, 192, 336, 720\}$ | $(12185, 1757, 3509)$ | 1 hour | Transportation |
| Short-term Forecasting | PeMS03 | 358 | 12 | $(15617, 5135, 5135)$ | 5 min | Traffic |
| | PeMS04 | 307 | 12 | $(10172, 3375, 3375)$ | 5 min | Traffic |
| | PeMS07 | 883 | 12 | $(16911, 5622, 5622)$ | 5 min | Traffic |
| | PeMS08 | 170 | 12 | $(10690, 3548, 265)$ | 5 min | Traffic |

We evaluate our method on eight established time series benchmarks for long-term forecasting. Additionally, we use the PeMS datasets for short-term forecasting. Dataset statistics are summarized in Table 5, with detailed descriptions provided below:

(1) The **ETT** (Electricity Transformer Temperature) dataset (Zhou et al., 2021) records temperature and load data from power transformers in two Chinese regions between 2016 and 2018. It includes two temporal resolutions: ETTh (hourly) and ETTm (15-minute intervals).

(2) The **Electricity** dataset (Wu et al., 2023) comprises hourly power consumption measurements (kWh) from 321 customers. Collected from the UCL repository and spanning 2012-2014, it captures residential and commercial energy usage patterns.

(3) The **Weather** dataset (Wu et al., 2023) contains 21 meteorological variables recorded at 10-minute intervals throughout 2020 in Germany. Parameters include temperature, humidity, pressure, and visibility, providing comprehensive environmental monitoring.

(4) The **Exchange** dataset (Wu et al., 2023) tracks daily currency values for eight major economies relative to the US dollar over 1990-2016. This 26-year series reflects global financial dynamics and macroeconomic trends.

(5) The **Traffic** dataset (Wu et al., 2023) provides hourly occupancy rates from 862 sensors on San Francisco Bay Area freeways during 2015-2016. It captures urban mobility patterns and congestion dynamics.

(6) The **PeMS** dataset (Chen et al., 2001) comprises four public traffic network datasets (PeMS03, PeMS04, PeMS07, PeMS08), collected from California's Caltrans Performance Measurement System (PeMS). The data from four distinct districts is aggregated at 5-minute intervals, yielding 12 observations per hour and 288 per day.

### B.4    MORE DETAILS OF EXPERIMENT

We make our codes publicly available, including implementations of all base models and the proposed AEA framework, to ensure reproducibility. The backbone implementations are adapted from their official GitHub repositories, with reference to the TimesNet codebase (Wu et al., 2023). All experiments were conducted using the following unified settings: batch size of 32, learning rate of 0.0005, random seeds of $\{2021, 2022, 2023\}$, and Adam optimizer (Kingma & Ba, 2015). Each run was trained for 10 epochs with early stopping (patience = 3) to prevent overfitting.

## C    MORE ANALYSIS

### C.1    MORE ANALYSIS OF HYPERPARAMETER SENSITIVITY

We conduct sensitivity analysis on four key hyperparameters using MSE as the evaluation metric, with DLinear as the backbone model under a forecasting horizon of 96.

**Sensitivity of Amplitude Mixing Ratio** $\alpha$**.**    We investigate the impact of the amplitude mixing ratio $\alpha$ in Spectral Mirroring, which controls the balance between the original and mirrored spectra amplitudes. As shown in Figure 3, performance remains stable across $\alpha \in [0.25, 0.75]$, with $\alpha = 0.5$ achieving optimal or near-optimal results on four datasets. This suggests that equal weighting offers the optimal balance between signal enhancement and distortion avoidance. The minimal performance variation ($< 6\%$ MSE difference across values) demonstrates the robustness of our amplitude mixing strategy to this hyperparameter.

**Sensitivity of Differential Scaling Initialization** $\lambda_{init}$**.**    The initialization of differential scaling parameters $\lambda_1$ and $\lambda_2$ is crucial for stable training. As shown in Figure 4, performance is largely insensitive to $\lambda_{init} \in [0, 1]$, with fluctuations within $5\%$ across datasets. The Softplus constraint ensures that positive values are maintained throughout the optimization process, while the learning mechanism allows for adaptation to dataset-specific noise characteristics. We use $\lambda_{init} = 0.2$ as the default for consistent convergence.

**Sensitivity of Non-stationarity Weight** $\lambda_{non-stat}$**.**    The regularization weight $\lambda_{non-stat}$ balances forecasting accuracy with representation stability. As shown in Figure 5, extreme values ($\geq 100$) cause noticeable degradation, while moderate settings ($0.1 - 1.0$) maintain stable performance. This confirms the importance of the non-stationarity loss for robust learning, while demonstrating that a wide range of values provides effective regularization. We set $\lambda_{non-stat} = 0.1$ as the default balanced configuration.

**Sensitivity of Energy Predictor Dimension** $D'$**.**    As shown in Figure 6, the embedding dimension $D'$ in the Energy Predictor shows minimal impact on performance, with differences $< 5\%$ across $D' \in [64, 512]$. This indicates that even compact representations ($D' = 64$) effectively capture the spectral mapping between enhanced and original distributions. The consistency across dimensions confirms the efficiency of our frequency-domain alignment approach.

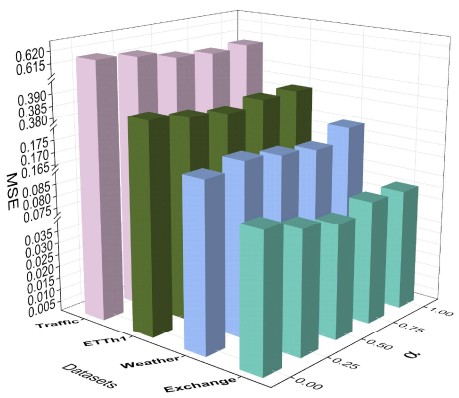

Figure 3: Performance *w.r.t.* different amplitude mixing ratio $\alpha$ with DLinear as backbone under a horizon of 96.

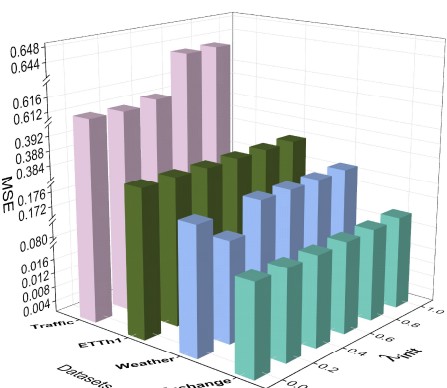

Figure 4: Performance *w.r.t.* different differential scaling initialization $\lambda_{init}$ with DLinear as backbone under a horizon of 96.

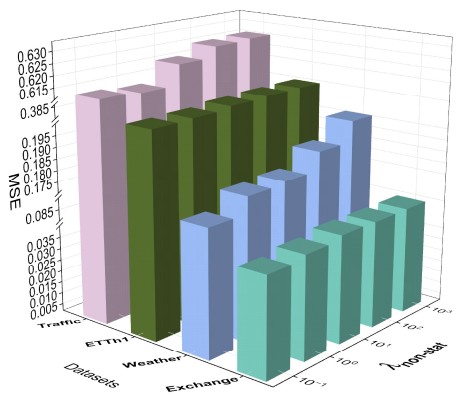

Figure 5: Performance *w.r.t.* different non-stationarity weight $\lambda_{non-stat}$ with DLinear as backbone under a horizon of 96.

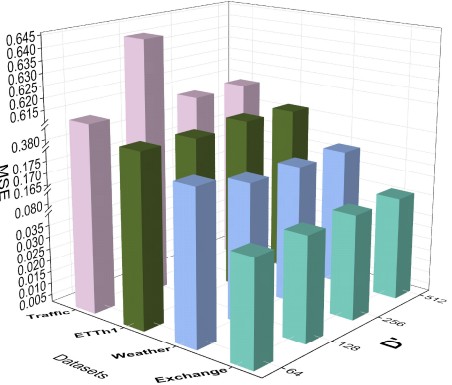

Figure 6: Performance *w.r.t.* different energy predictor dimension $D'$ with DLinear as backbone under a horizon of 96.

Table 6: Ablation study results across five datasets. Models are compared in terms of MSE and MAE (lower values are better) using the **PatchTST** backbone under a forecasting horizon of 96. The best result for each dataset is highlighted in **bold**. 'Avg' denotes the average results of MSE and MAE. The last column, 'Drop (%)', shows the average performance deterioration percentage of all datasets.

|  | ETTh1 | ETTh2 | Weather | Exchange | Traffic | Avg | Drop (%) |
|---|---|---|---|---|---|---|---|
| **AEA** | **0.385** | **0.314** | **0.196** | **0.144** | **0.394** | **0.287** | - |
| *w/o Spectral Mirroring* | 0.399 | 0.322 | 0.220 | 0.152 | 0.406 | 0.300 | 6.656 |
| *w/o Phase Mixing* | 0.403 | 0.315 | 0.207 | 0.157 | 0.394 | 0.295 | 4.993 |
| *w/o Differential Embedding* | 0.399 | 0.326 | 0.224 | 0.161 | 0.416 | 0.305 | 9.741 |
| *w/o Non-stationarity Loss* | 0.393 | 0.324 | 0.215 | 0.152 | 0.414 | 0.299 | 6.399 |
| *w/o Energy Predictor* | 0.415 | 0.346 | 0.242 | 0.145 | 0.411 | 0.312 | 11.247 |

## C.2 ABLATION STUDY ON NON-LINEAR BACKBONE.

To verify the generality of the module contributions, we conducted an ablation study using the non-linear PatchTST backbone. The results, presented in Table 6, strongly align with our initial findings on DLinear (refer to Section 5.4). The consistent performance hierarchy shows that the Energy Predictor and Differential Embedding remain the most critical components, indicating that their functional roles are fundamental to the framework's operation. This cross-architectural consistency

Table 7: Performance comparison of adaptive *v.s.* fixed $\alpha$ in AEA. The forecasting horizons are $\{96, 192, 336, 720\}$. The better performance in each setting is shown in **bold**. 'Avg' denotes the average results of four forecasting horizons.

| Model Variant Metric | | DLinear + AEA | | | | PatchTST + AEA | | | |
|---|---|---|---|---|---|---|---|---|---|
| | | Adaptive | | Fixed | | Adaptive | | Fixed | |
| | | MSE | MAE | MSE | MAE | MSE | MAE | MSE | MAE |
| ETTh1 | 96 | 0.380 | 0.392 | 0.380 | 0.392 | 0.381 | 0.399 | **0.374** | **0.396** |
| | 192 | 0.432 | 0.427 | 0.432 | **0.422** | 0.449 | 0.438 | **0.428** | **0.431** |
| | 336 | **0.474** | **0.450** | 0.479 | 0.458 | 0.481 | 0.455 | **0.466** | **0.452** |
| | 720 | 0.496 | 0.490 | **0.495** | **0.489** | 0.557 | 0.500 | **0.479** | **0.478** |
| | Avg | **0.445** | 0.440 | 0.447 | 0.440 | 0.467 | 0.448 | **0.437** | **0.439** |
| ETTh2 | 96 | 0.330 | 0.385 | **0.329** | **0.384** | 0.291 | 0.340 | **0.290** | **0.339** |
| | 192 | 0.453 | 0.461 | **0.450** | **0.457** | 0.377 | **0.391** | **0.375** | 0.392 |
| | 336 | 0.577 | **0.531** | **0.576** | 0.533 | **0.415** | 0.425 | 0.416 | 0.425 |
| | 720 | 0.804 | 0.644 | **0.795** | **0.641** | 0.433 | 0.447 | **0.421** | **0.439** |
| | Avg | 0.541 | 0.505 | **0.538** | **0.504** | 0.379 | 0.401 | **0.375** | **0.399** |
| Exchange | 96 | 0.078 | 0.201 | **0.077** | **0.199** | 0.088 | 0.206 | **0.084** | **0.204** |
| | 192 | 0.158 | **0.294** | 0.158 | 0.295 | 0.195 | 0.313 | **0.186** | **0.306** |
| | 336 | **0.266** | **0.389** | 0.270 | 0.394 | 0.416 | 0.477 | **0.323** | **0.410** |
| | 720 | 0.792 | 0.674 | **0.773** | **0.667** | **0.861** | **0.694** | 0.877 | 0.707 |
| | Avg | 0.323 | 0.389 | **0.320** | 0.389 | 0.390 | 0.422 | **0.367** | **0.407** |
| Weather | 96 | 0.172 | 0.246 | **0.169** | **0.244** | 0.178 | 0.226 | **0.173** | **0.219** |
| | 192 | 0.217 | 0.288 | **0.213** | **0.286** | **0.217** | 0.256 | 0.218 | 0.256 |
| | 336 | 0.268 | 0.329 | **0.266** | **0.327** | **0.272** | **0.295** | 0.273 | 0.296 |
| | 720 | 0.338 | 0.380 | **0.336** | **0.378** | **0.350** | 0.347 | 0.351 | **0.345** |
| | Avg | 0.249 | 0.311 | **0.246** | **0.309** | 0.254 | 0.281 | 0.254 | **0.279** |
| Traffic | 96 | 0.614 | 0.399 | **0.611** | **0.388** | 0.481 | **0.295** | 0.481 | 0.306 |
| | 192 | 0.594 | 0.371 | **0.589** | **0.370** | 0.488 | **0.297** | **0.486** | 0.301 |
| | 336 | 0.600 | **0.372** | **0.597** | 0.374 | 0.505 | **0.300** | **0.503** | 0.309 |
| | 720 | 0.640 | 0.393 | **0.634** | **0.392** | **0.535** | **0.320** | 0.541 | 0.328 |
| | Avg | 0.612 | 0.384 | **0.608** | **0.381** | **0.502** | **0.303** | 0.503 | 0.311 |
| **1st Count** | | 3 | 5 | 19 | 17 | 7 | 8 | 16 | 15 |

underscores that AEA provides a robust and model-agnostic enhancement for time series forecasting.

## C.3   ADDITIONAL DISCUSSION OF ADAPTIVE AMPLITUDE MIXING RATIO

The amplitude mixing ratio $\alpha$ balances the original and mirrored spectra during the Spectral Mirroring process. To examine its flexibility, we compared our fixed $\alpha=0.5$ setting against a learnable $\alpha$ parameter, with results provided in Table 7. The fixed ratio achieves superior performance in the majority of settings (*e.g.*, 19 *v.s.* 3 best scores for DLinear and 16 *v.s.* 7 for PatchTST). This consistent advantage demonstrates that a fixed, balanced mixture provides a more effective and stable foundation for subsequent processing. The necessary adaptation to specific frequency characteristics is then more efficiently handled by the subsequent learnable scaling matrix and the Differential Embedding module, which perform fine-grained adjustments. Consequently, we retain the fixed $\alpha=0.5$ for its proven effectiveness.

Table 8: Standard deviation and statistical tests for Amplifier and FredFormer, with their counterparts enhanced by the AEA. The results are averaged by four forecasting horizons $\{96, 192, 336, 720\}$.

| Model | Amplifier | | Amplifier + AEA | | Confidence | FredFormer | | FredFormer + AEA | | Confidence |
|-------|-----|-----|-----|-----|-----|-----|-----|-----|-----|-----|
| Dataset | MSE | MAE | MSE | MAE | Level | MSE | MAE | MSE | MAE | Level |
| ETTh1 | $0.473 \pm 0.011$ | $0.454 \pm 0.009$ | $0.452 \pm 0.014$ | $0.439 \pm 0.008$ | 0.99 | $0.451 \pm 0.005$ | $0.437 \pm 0.009$ | $0.443 \pm 0.009$ | $0.427 \pm 0.012$ | 0.99 |
| ETTh2 | $0.382 \pm 0.019$ | $0.407 \pm 0.017$ | $0.374 \pm 0.008$ | $0.400 \pm 0.011$ | 0.99 | $0.365 \pm 0.005$ | $0.394 \pm 0.008$ | $0.358 \pm 0.014$ | $0.389 \pm 0.014$ | 0.99 |
| ETTm1 | $0.390 \pm 0.015$ | $0.398 \pm 0.012$ | $0.387 \pm 0.009$ | $0.395 \pm 0.009$ | 0.99 | $0.403 \pm 0.007$ | $0.404 \pm 0.010$ | $0.391 \pm 0.012$ | $0.377 \pm 0.016$ | 0.99 |
| ETTm2 | $0.278 \pm 0.006$ | $0.324 \pm 0.014$ | $0.276 \pm 0.005$ | $0.323 \pm 0.004$ | 0.99 | $0.282 \pm 0.011$ | $0.326 \pm 0.006$ | $0.279 \pm 0.014$ | $0.324 \pm 0.009$ | 0.99 |
| Electricity | $0.283 \pm 0.008$ | $0.328 \pm 0.006$ | $0.279 \pm 0.009$ | $0.326 \pm 0.014$ | 0.99 | $0.178 \pm 0.004$ | $0.270 \pm 0.015$ | $0.174 \pm 0.016$ | $0.266 \pm 0.011$ | 0.99 |
| Exchange | $0.369 \pm 0.018$ | $0.406 \pm 0.010$ | $0.360 \pm 0.010$ | $0.402 \pm 0.006$ | 0.99 | $0.354 \pm 0.010$ | $0.400 \pm 0.014$ | $0.348 \pm 0.013$ | $0.391 \pm 0.016$ | 0.99 |
| Weather | $0.255 \pm 0.009$ | $0.279 \pm 0.015$ | $0.252 \pm 0.011$ | $0.277 \pm 0.008$ | 0.99 | $0.245 \pm 0.011$ | $0.272 \pm 0.009$ | $0.243 \pm 0.009$ | $0.270 \pm 0.008$ | 0.99 |
| Traffic | $0.561 \pm 0.012$ | $0.360 \pm 0.006$ | $0.543 \pm 0.014$ | $0.348 \pm 0.009$ | 0.99 | $0.445 \pm 0.011$ | $0.300 \pm 0.021$ | $0.440 \pm 0.017$ | $0.293 \pm 0.019$ | 0.99 |

Table 9: Efficiency analysis of FredFormer and its AEA-enhanced variant on the Electricity dataset across forecasting horizons $\{96, 192, 336, 720\}$. Metrics include: training time (s/epoch), inference time (s/iter), FLOPs (G), measuring computational complexity; MACs (M), indicating hardware performance requirements; and number of parameters, representing model size.

| Dataset | Horizon | FredFormer | | | | | FredFormer + AEA | | | | |
|---------|---------|-----|-----|-----|-----|-----|-----|-----|-----|-----|-----|
| | | train time (s/epoch) | infer time (s/iter) | FLOPs (G) | MACs (M) | parameters | train time (s/epoch) | infer time (s/iter) | FLOPs (G) | MACs (M) | parameters |
| Electricity | 96 | 40.103 | 0.040 | 177.299 | 11.866 | 12116801 | 52.823 | 0.041 | 177.452 | 11.875 | 12141511 |
| | 192 | 41.023 | 0.053 | 179.886 | 12.118 | 12516641 | 52.555 | 0.053 | 180.191 | 12.142 | 12556183 |
| | 336 | 42.445 | 0.066 | 184.478 | 12.565 | 13358321 | 53.788 | 0.075 | 185.122 | 12.622 | 13431007 |
| | 720 | 42.445 | 0.066 | 200.887 | 14.164 | 17022065 | 59.435 | 0.088 | 203.368 | 14.400 | 17273887 |
| | Avg. | 41.504 | 0.056 | 185.638 | 12.678 | 13753457 | 54.650 | 0.064 | 186.533 | 12.760 | 13850647 |

Table 10: Efficiency analysis of Amplifier and its AEA-enhanced variant on the Electricity dataset across forecasting horizons $\{96, 192, 336, 720\}$. Metrics include: training time (s/epoch), inference time (s/iter), FLOPs (G), measuring computational complexity; MACs (M), indicating hardware performance requirements; and number of parameters, representing model size.

| Dataset | Horizon | Amplifier | | | | | Amplifier + AEA | | | | |
|---------|---------|-----|-----|-----|-----|-----|-----|-----|-----|-----|-----|
| | | train time (s/epoch) | infer time (s/iter) | FLOPs (G) | MACs (M) | parameters | train time (s/epoch) | infer time (s/iter) | FLOPs (G) | MACs (M) | parameters |
| Electricity | 96 | 14.236 | 0.021 | 1.688 | 0.502 | 518153 | 22.055 | 0.030 | 1.752 | 0.508 | 524684 |
| | 192 | 15.316 | 0.023 | 2.205 | 0.603 | 619049 | 22.861 | 0.029 | 2.333 | 0.622 | 638012 |
| | 336 | 19.126 | 0.028 | 5.759 | 1.395 | 1411417 | 22.807 | 0.036 | 6.038 | 1.444 | 1459924 |
| | 720 | 25.544 | 0.053 | 9.846 | 2.192 | 2208217 | 30.477 | 0.055 | 10.996 | 2.410 | 2426260 |
| | Avg. | 18.555 | 0.031 | 4.875 | 1.173 | 1189209 | 24.550 | 0.038 | 5.280 | 1.246 | 1262220 |

## C.4 ERROR BARS

In this paper, we repeat all the experiments three times. Here we report the standard deviation of Amplifier and FredFormer, as well as the statistical significance test in Table 8.

## C.5 INTUITION OF USING LOW-FREQUENCY SPECTRUM TO GUIDE HIGH FREQUENCY

The rationale is directly derived from the **spectral energy imbalance** of time series data. As established via Parseval's Theorem and observed in real-world time series, the signal energy is overwhelmingly concentrated in the low-frequency components due to their high amplitudes. This high amplitude directly translates to a higher inherent Signal-to-Noise Ratio (SNR): the powerful low-frequency signal is far less susceptible to being corrupted or obscured by background noise compared to the subtle, low-amplitude high-frequency components, which can be easily drowned out by stochastic noise. It is this inherent robustness that qualifies low-frequency components as a reliable source for constructing a stable template. Crucially, this guidance is not rigid but is adaptively controlled by the learnable scaling matrix and refined by the Differential Embedding module, which is empirically validated by the consistent improvements across diverse datasets.

## C.6 EFFICIENCY ANALYSIS

To assess the computational overhead of AEA, we conducted efficiency experiments on the Electricity dataset with 321 variables across four forecasting horizons $\{96, 192, 336, 720\}$. The results in Table 9 and 10 demonstrate that AEA introduces minimal practical cost while delivering significant performance improvements. For the FredFormer backbone, AEA increases average training time per epoch by 13 seconds and inference time by only 0.008 seconds, with computational metrics showing negligible increases of 0.483% in FLOPs and 0.707% in parameters. The Amplifier backbone shows slightly higher relative increases in computational metrics due to its simpler base architecture, yet

maintains practically insignificant absolute overhead with just 6 seconds additional training time per epoch and 0.006 seconds in inference. These findings confirm that AEA's components are efficiently implemented, making the framework suitable for real-world deployment even with high-dimensional data and long forecasting horizons.

Table 11: Long-term forecasting on Electricity dataset with different look-back window length in $\{96, 192, 336, 720\}$. The forecasting lengths are $\{96, 192, 336, 720\}$. **Bold** means AEA successfully enhances forecasting performance over the base model.

| Window Length | Horizon | Amplifier | | Amplifier + AEA | |
|---|---|---|---|---|---|
| | | mse | mae | mse | mae |
| 96 | 96 | 0.178 | 0.267 | **0.173** | **0.265** |
| | 192 | 0.247 | 0.306 | **0.243** | **0.303** |
| | 336 | 0.308 | 0.343 | **0.302** | **0.340** |
| | 720 | 0.398 | 0.396 | **0.397** | 0.396 |
| | Avg. | 0.283 | 0.328 | **0.279** | **0.326** |
| 192 | 96 | 0.145 | 0.239 | **0.139** | **0.236** |
| | 192 | 0.161 | 0.252 | **0.157** | 0.252 |
| | 336 | 0.178 | 0.272 | **0.174** | **0.268** |
| | 720 | 0.215 | 0.301 | **0.193** | **0.284** |
| | Avg. | 0.175 | 0.266 | **0.166** | **0.260** |
| 336 | 96 | 0.141 | 0.238 | **0.139** | **0.235** |
| | 192 | 0.160 | 0.253 | **0.155** | **0.248** |
| | 336 | 0.174 | 0.270 | **0.167** | **0.259** |
| | 720 | 0.205 | 0.293 | **0.202** | **0.291** |
| | Avg. | 0.170 | 0.264 | **0.166** | **0.258** |
| 720 | 96 | 0.139 | 0.238 | **0.137** | **0.234** |
| | 192 | 0.153 | 0.249 | **0.150** | **0.246** |
| | 336 | 0.178 | 0.277 | **0.162** | **0.258** |
| | 720 | 0.195 | 0.286 | **0.191** | **0.284** |
| | Avg. | 0.166 | 0.263 | **0.160** | **0.255** |

## C.7 INFLUENCE OF LOOK-BACK WINDOW LENGTH

In this section, we investigate the effect of different lookback window lengths $\{96, 192, 336, 720\}$ on the Electricity dataset. We evaluate window sizes of $\{96, 192, 336, 720\}$ to understand how historical context length influences model behavior and to identify potential risks of overfitting with excessively long windows or underfitting with insufficient historical context. As shown in Table 11, AEA consistently enhances base model performance across all forecasting horizons and window lengths. The framework demonstrates particular effectiveness with a 192-step lookback window, suggesting this length provides an optimal balance between capturing sufficient temporal dependencies and avoiding overfitting.

## C.8 VISUAL ANALYSIS

Visual analysis of forecasting results on the Electricity dataset (horizon=96) highlights the distinct and complementary improvements enabled by the AEA framework. As shown in Figures 7 and 8, AEA enhances Amplifier's capacity to capture fine-grained temporal variations, resulting in predictions with **sharper and more accurate peak representations**. This indicates that our method effectively

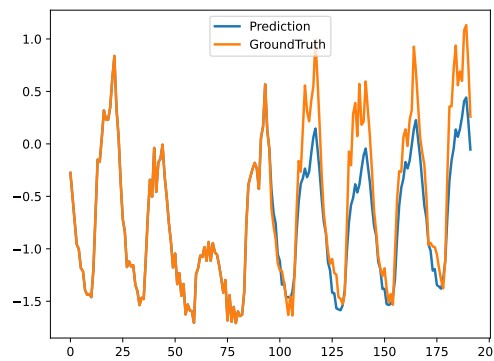

Figure 7: Forecasting visualization of Amplifier on the Electricity dataset under a horizon of 96.

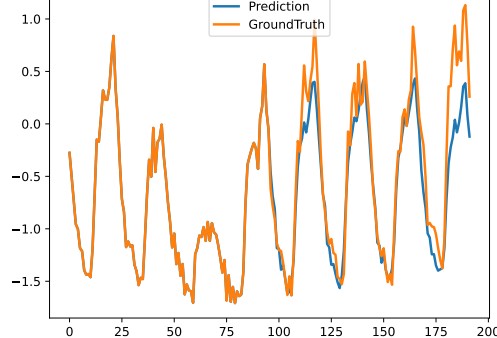

Figure 8: Forecasting visualization of Amplifier + AEA on the Electricity dataset under a horizon of 96.

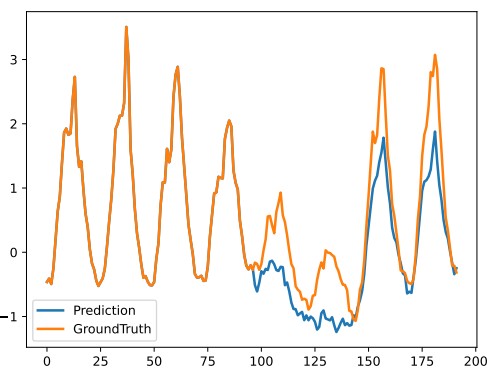

Figure 9: Forecasting visualization of FredFormer on the Electricity dataset under a horizon of 96.

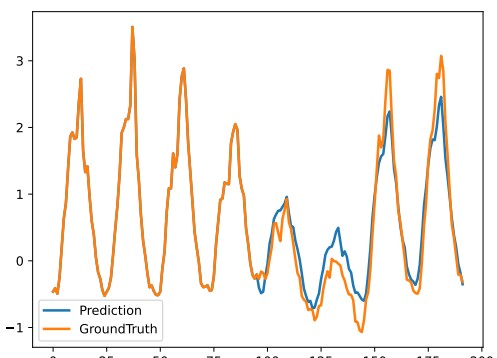

Figure 10: Forecasting visualization of FredFormer + AEA on the Electricity dataset under a horizon of 96.

mitigates spectral bias, allowing the model to better model rapid signal changes. Meanwhile, Figures 9 and 10 reveal that for **FredFormer**, AEA primarily contributes to output stabilization by alleviating temporal distribution shift and suppressing spurious fluctuations, yielding a smoother and more consistent forecast trajectory. These findings demonstrate AEA's ability to provide robust, model-agnostic enhancement through adaptive frequency amplification and noise suppression.

## D  LIMITATIONS AND FUTURE WORK

The proposed AEA framework has demonstrated its effectiveness as a model-agnostic enhancement across multiple forecasting backbones, including linear models, Transformers, convolutional architectures, and specialized frequency-domain methods. However, our current evaluation focuses primarily on these conventional deep learning architectures. This study does not include the emerging class of LLM-based time series models, such as Time-LLM (Jin et al., 2024) and AutoTimes (Liu et al., 2024b). This limitation arises from the substantial computational requirements of these billion-scale parameter models and their distinct architectural paradigms, which present unique integration challenges beyond the scope of this initial investigation. Future work will explore adapting AEA's core principles of spectral bias mitigation to LLM-based forecasting architectures. The mechanisms of adaptive energy amplification and noise suppression could potentially enhance how these foundation models capture predictive high-frequency signals, opening promising new research directions in time series forecasting.

