# OpenReview forum: "Adaptive Energy Amplification for Robust Time Series Forecasting"
_ICLR.cc/2026/Conference — Submitted to ICLR 2026_

### Official Review · Reviewer_G4Y1 · 2025-10-29

**Soundness:** 2
**Presentation:** 3
**Contribution:** 2
**Rating:** 4
**Confidence:** 4

**Summary:**

This paper addresses the pervasive issue of spectral bias in deep learning models for time series forecasting, where models tend to overfit low-frequency, high-energy signals and neglect potentially predictive yet low-energy high-frequency components. The authors propose Adaptive Energy Amplification (AEA), a model-agnostic frequency-domain framework that (1) employs spectral mirroring for phase-preserving, distortion-free amplification of high-frequency signals based on low-frequency surrogates, and (2) integrates a lightweight differential embedding designed to suppress common-mode noise in the latent space. Extensive experiments across eight real-world datasets and four prominent forecasting backbones demonstrate that AEA not only improves forecasting accuracy but also training stability and robustness to high-frequency noise.

**Strengths:**

1.	The paper rigorously identifies indiscriminate amplification as a key limitation of prior frequency-domain approaches and provides empirical evidence motivating selective, content-aware high-frequency enhancement. The proposed AEA module is formulated as a lightweight plug-in that can be broadly applied to existing forecasting architectures.
2.	AEA is presented with detailed mathematical exposition and clear component decomposition. Fig. 2 effectively illustrates the conceptual and operational flow among spectral mirroring, differential embedding, and spectral energy alignment, enabling readers to understand both the intuition and implementation pathway.

**Weaknesses:**

1.	The proposed Spectral Mirroring + Adaptive Scaling framework closely resembles Amplifier. The additional components—phase mixing, differential embedding, and non-stationarity regularization—appear incremental rather than conceptually groundbreaking.
2.	The paper claims that existing models neglect high-frequency components. However, Fig. 1(a) shows that masking high frequencies leads to only minor degradation (< 5%), whereas masking low frequencies causes > 100% loss. This observation undermines the stated motivation and weakens the justification for “energy amplification” of high-frequency bands.
3.	Average improvements across datasets are around 3–4% (sometimes < 1%) in MAE/MSE. The paper reports win ratios instead of statistical tests, making it difficult to judge whether these gains are statistically meaningful or merely within random variation.
4.	The reproduced Amplifier results are significantly higher than those originally reported. Given that AEA’s claimed improvements are largely measured against this weakened baseline, the validity of the comparative conclusions is questionable.
5.	Although AEA is claimed to be model-agnostic, experiments are limited to DLinear, PatchTST, TimesNet, and Amplifier. Comparisons with stronger spectral models (e.g., FEDformer, FITS, FreTS) and emerging LLM-based forecasting architectures (e.g., AutoTimes, Time-LLM) are missing, restricting the assessment of AEA’s competitiveness.
6.	The paper omits efficiency profiling in comparison with baselines. Since AEA adds rFFT/iFFT, phase mixing, differential embeddings, and an energy predictor, the end-to-end cost may be non-trivial—particularly for long horizons and high-dimensional multivariate series.
7.	The paper would benefit from qualitative visualizations (e.g., input/output spectra or time-domain signals) illustrating what high-frequency structures are amplified and what types of noise are suppressed, thereby enhancing interpretability.
8.	It remains unclear whether learning dataset- or task-specific mixing ratios (instead of a fixed $\alpha$) could yield more adaptive behavior and improved performance.

**Questions:**

See weaknesses.

---

> ### Author Response · Authors · 2025-11-28
> **Response to Reviewer G4Y1 (Part 1)**
>
> ## Weakness 1: The conceptual advancement beyond Amplifier appears incremental.
>
> Thank you for this comment regarding the conceptual contribution. While Spectral Mirroring shares the high-level idea of spectrum reversal with Amplifier, our work introduces a fundamental **paradigm shift** from *indiscriminate amplification* to *adaptive enhancement*.
>
> The core limitation of Amplifier is its **indiscriminate amplification** of all high-frequency components, which inevitably elevates noise alongside signals. Our framework addresses this through three principled innovations that collectively solve a different problem:
>
> 1.  **Phase Mixing** ensures the enhancement process is **distortion-free**, preserving the temporal fidelity of the reconstructed signal, which is a critical aspect overlooked in prior work.
> 2.  **Differential Embedding** provides the crucial capability of **adaptive noise suppression**, actively filtering out common-mode noise that would otherwise be amplified.
> 3.  **Non-stationarity Loss** stabilizes the learning process against batch-wise variations, enabling robust training.
>
> These components are not incremental additions but form a **cohesive, interdependent system** that solves the fundamental problem of *discriminative amplification*. The empirical evidence strongly supports this: AEA not only consistently outperforms Amplifier (Table 1) but, more importantly, maintains its performance advantage even when high-frequency noise is artificially injected (Fig. 1b), demonstrating its unique robustness.
>
> Therefore, our contribution lies not in individual components but in a **novel framework that reframes the problem and provides a principled, robust solution** to the critical limitation of indiscriminate amplification.
>
> ## Weakness 2: The motivation is seemingly contradicted by frequency masking results.
>
> Thank you for this comment. The observation from Figure 1(a) is consistent with and strongly supports our motivation.
>
> The fact that masking high frequencies causes only minor performance loss clearly illustrates the spectral bias problem: current models have learned to rely predominantly on low-frequency information and consequently neglect the predictive potential within high-frequency bands. This does not imply that high-frequency components are unimportant, but rather that they are severely under-utilized by existing models due to the natural energy imbalance during training.
>
> Our method addresses this by recalibrating the learning process through adaptive energy amplification. This guides the model to discover and leverage the valuable signals within high-frequency components. The consistent performance gains achieved by AEA across all benchmarks confirm that these high-frequency signals are indeed highly predictive once the model is properly directed to attend to them.
>
> ## Weakness 3: The statistical significance of performance gains is not established.
>
> Thank you for raising the important point about statistical significance. We have now added comprehensive statistical analysis in the new **Table 8**, which provides standard deviations and significance tests across all datasets.
>
> The results consistently show statistically significant improvements at the 99% confidence level for both Amplifier+AEA and FredFormer+AEA compared to their base models. While the average improvements of 3-4% may appear modest, they represent consistent gains across eight diverse benchmarks and multiple backbones. In the context of competitive forecasting benchmarks where state-of-the-art models are already highly optimized, such consistent improvements are both statistically significant and practically meaningful. The statistical evidence now provided in Table 8 confirms that these gains are beyond random variation and demonstrate the robust effectiveness of our approach.
>
> Table 8: Standard deviation and statistical tests for Amplifier and FredFormer. More Results are in Appendix C.4.
>
> | Dataset | Amplifier |  | Amplifier + AEA |  | Confidence Level | FredFormer |  | FredFormer + AEA |  | Confidence Level |
> |:---:|:---:|:---:|:---:|:---:|:---:|:---:|:---:|:---:|:---:|:---:|
> |  | mse | mse std | mse | mse std |  | mse | mse std | mse | mse std |  |
> | ETTh1 | 0.473 | 0.011 | 0.452 | 0.014 | 99% | 0.451 | 0.005 | 0.443 | 0.009 | 99% |
> | ETTh2 | 0.382 | 0.019 | 0.374 | 0.008 | 99% | 0.365 | 0.005 | 0.358 | 0.014 | 99% |

---

> > ### Author Response · Authors · 2025-11-28
> > **Response to Reviewer G4Y1 (Part 2)**
> >
> > ## Weakness 4: Reproduced results for the Amplifier baseline differ from previously reported values.
> >
> > Thank you for this important observation. The performance difference in Amplifier stems from our implementation within a unified experimental framework (refer to TimesNet [1]) to ensure strictly controlled comparisons across all models. While this affected absolute performance metrics, it does not compromise the validity of our comparative conclusions for two key reasons.
> >
> > First, the relative improvements brought by AEA are measured consistently within this controlled environment. The significant and consistent performance gains of AEA+Amplifier over our implemented Amplifier baseline (demonstrated in Table 1 and statistically validated in Table 8) remain meaningful, showing that AEA provides substantial enhancement regardless of the baseline's absolute performance.
> >
> > Second, and more importantly, AEA's effectiveness is demonstrated universally across all six backbones, not just Amplifier. The framework consistently improves models with varying initial performances in our setup, which strongly supports its general utility as a model-agnostic enhancement rather than an artifact of any particular baseline implementation.
> >
> > The consistent improvements across this diverse model spectrum, all trained and evaluated under the same rigorous protocol, provide robust evidence for AEA's effectiveness.
> >
> > [1] Wu et al. TimesNet: Temporal 2D-Variation Modeling for General Time Series Analysis. ICLR 2023.
> >
> > ## Weakness 5：Model comparisons lack recent frequency-aware and LLM-based baselines.
> >
> > Thank you for this valuable suggestion. We agree that evaluating AEA against a broader range of architectures is important. In response, we have significantly expanded our experiments to include two additional SOTA frequency-domain models, **FreTS** and **FredFormer**, in Table 1. The results show that AEA provides consistent improvements for these specialized spectral methods as well, reinforcing its general applicability.
> >
> > Regarding LLM-based forecasters (e.g., Time-LLM, AutoTimes), we agree that exploring their integration with AEA is a fascinating direction. However, these models typically possess billions of parameters, and their training and inference require substantial computational resources that fall outside the scope and capacity of our current experimental setup.
> >
> > Crucially, this computational constraint does not reflect a limitation of the AEA framework itself. The core mechanisms of AEA—operating on the frequency spectrum of the input data—are fundamentally compatible with any time series forecasting architecture. The framework is designed to be a plug-and-play module that can, in principle, be integrated into the preprocessing or feature extraction stage of an LLM-based forecaster.
> >
> > We have added a discussion of this limitation and the promising direction of adapting AEA for LLM-based forecasters as future work. We believe the expanded evaluation across six diverse backbones—spanning linear, transformer, convolutional, and frequency-domain paradigms—now provides comprehensive evidence of AEA's general applicability and competitiveness.
> >
> > Table 1: Long-term forecasting performance in the etth1 dataset. More results are in Section 5.2.
> >
> > | Dataset | Setting | FreTS |  | FreTS + AEA |  | FredFormer |  | FredFormer + AEA |  |
> > |:---:|:---:|:---:|:---:|:---:|:---:|:---:|:---:|:---:|:---:|
> > |  |  | mse | mae | mse | mae | mse | mae | mse | mae |
> > | etth1 | 96 | 0.395  | 0.406  | 0.386  | 0.400  | 0.376  | 0.395  | 0.375  | 0.393  |
> > |  | 192 | 0.454  | 0.443  | 0.441  | 0.436  | 0.438  | 0.427  | 0.434  | 0.426  |
> > |  | 336 | 0.517  | 0.484  | 0.489  | 0.459  | 0.485  | 0.448  | 0.484  | 0.446  |
> > |  | 720 | 0.589  | 0.536  | 0.529  | 0.513  | 0.506  | 0.479  | 0.479  | 0.441  |
> > |  | Avg. | 0.488  | 0.467  | 0.461  | 0.452  | 0.451  | 0.437  | 0.443  | 0.427  |

---

> > > ### Author Response · Authors · 2025-11-28
> > > **Response to Reviewer G4Y1 (Part 3)**
> > >
> > > ## Weakness 6: Computational efficiency is not profiled.
> > >
> > > Thank you for raising this important point regarding computational efficiency. We have conducted a comprehensive efficiency analysis on the Electricity dataset (321 variates) across forecasting horizons {96, 192, 336, 720}, with results detailed in **Tables 9 and 10**.
> > >
> > > The empirical measurements demonstrate that AEA introduces minimal computational overhead even on this high-dimensional dataset. For the transformer-based FredFormer (Table 9), AEA increases average training time by 13 seconds while adding only 0.008 seconds to inference time. The computational footprint remains modest with FLOPs increasing by only 0.483%, MACs by 0.643%, and parameters by 0.707%.
> > >
> > > For the linear architecture Amplifier (Table 10), the relative increases in computational metrics are higher (FLOPs: 8.314%, MACs: 6.235%, parameters: 6.139%) due to the simpler base model, yet the absolute overhead remains practically insignificant - just 6 seconds additional training time and 0.006 seconds inference time on average.
> > >
> > > These results confirm that AEA's components—including rFFT/iFFT, phase mixing, differential embeddings, and energy predictor—are efficiently implemented. The framework maintains practical computational efficiency while providing substantial performance improvements, making it suitable for real-world deployment with high-dimensional multivariate series and long forecasting horizons. We have added this complete efficiency analysis to **Appendix C.6** of our revised manuscript.
> > >
> > > **Table 9: Efficiency analysis of FredFormer and its AEA-enhanced variant on the Electricity dataset.**
> > > | Dataset | Setting | FredFormer |  |  |  |  | FredFormer + AEA |  |  |  |  |
> > > |:---:|:---:|:---:|:---:|:---:|:---:|:---:|:---:|:---:|:---:|:---:|:---:|
> > > |  |  | train time (s/epoch) | infer time (s/iter) | FLOPs (G) | MACs (M) | parameters | train time (s/epoch) | infer time (s/iter) | FLOPs (G) | MACs (M) | parameters |
> > > | Electricity | 96 | 40.103  | 0.040  | 177.299  | 11.866  | 12116801  | 52.823  | 0.041  | 177.452  | 11.875  | 12141511  |
> > > |  | 192 | 41.023  | 0.053  | 179.886  | 12.118  | 12516641  | 52.555  | 0.053  | 180.191  | 12.142  | 12556183  |
> > > |  | 336 | 42.445  | 0.066  | 184.478  | 12.565  | 13358321  | 53.788  | 0.075  | 185.122  | 12.622  | 13431007  |
> > > |  | 720 | 42.445  | 0.066  | 200.887  | 14.164  | 17022065  | 59.435  | 0.088  | 203.368  | 14.400  | 17273887  |
> > > |  | Avg. | 41.504  | 0.056  | 185.638  | 12.678  | 13753457  | 54.650  | 0.064  | 186.533  | 12.760  | 13850647  |
> > >
> > > **Table 10: Efficiency analysis of Amplifier and its AEA-enhanced variant on the Electricity dataset.**
> > > | Dataset | Setting | Amplifier |  |  |  |  | Amplifier + AEA |  |  |  |  |
> > > |:---:|:---:|:---:|:---:|:---:|:---:|:---:|:---:|:---:|:---:|:---:|:---:|
> > > |  |  | train time (s/epoch) | infer time (s/iter) | FLOPs (G) | MACs (M) | param | train time (s/epoch) | infer time (s/iter) | FLOPs (G) | MACs (M) | param |
> > > | Electricity | 96 | 14.236  | 0.021  | 1.688  | 0.502  | 518153  | 22.055  | 0.030  | 1.752  | 0.508  | 524684  |
> > > |  | 192 | 15.316  | 0.023  | 2.205  | 0.603  | 619049  | 22.861  | 0.029  | 2.333  | 0.622  | 638012  |
> > > |  | 336 | 19.126  | 0.028  | 5.759  | 1.395  | 1411417  | 22.807  | 0.036  | 6.038  | 1.444  | 1459924  |
> > > |  | 720 | 25.544  | 0.053  | 9.846  | 2.192  | 2208217  | 30.477  | 0.055  | 10.996  | 2.410  | 2426260  |
> > > |  | Avg. | 18.555  | 0.031  | 4.875  | 1.173  | 1189209  | 24.550  | 0.038  | 5.280  | 1.246  | 1262220  |
> > >
> > > ## Weakness 7: Qualitative analysis and visual evidence are missing.
> > >
> > > We present a visual analysis in **Appendix C.8**. Visual analysis of forecasting results on the Electricity dataset (horizon=96) highlights the distinct and complementary improvements enabled by the AEA framework. As shown in **Figures 7 and 8**, AEA enhances **Amplifier**'s capacity to capture fine-grained temporal variations, resulting in predictions with **sharper and more accurate peak representations**. This indicates that our method effectively mitigates spectral bias, allowing the model to better model rapid signal changes. Meanwhile, **Figures 9 and 10** reveal that for **FredFormer**, AEA primarily contributes to output stabilization by alleviating **temporal distribution shift** and suppressing spurious fluctuations, yielding a smoother and more consistent forecast trajectory. Together, these visual examples illustrate how AEA’s dual mechanism of adaptive frequency amplification and noise suppression delivers robust and model-agnostic performance enhancements, tailored to the specific characteristics of different forecasting backbones.

---

> > > > ### Author Response · Authors · 2025-11-28
> > > > **Response to Reviewer G4Y1 (Part 4)**
> > > >
> > > > ## Weakness 8: The fixed mixing ratio requires stronger justification.
> > > >
> > > > Thank you for raising this point regarding the amplitude mixing ratio $\alpha$. Our choice of $\alpha$=0.5 is not arbitrary but was determined through systematic hyperparameter sensitivity analysis, making it an empirically optimized parameter rather than a rigid default. We have conducted additional experiments comparing this fixed, optimized value against a fully learnable $\alpha$ variant (refer to **Table 7**). The results consistently show that the fixed ratio achieves comparable or better performance across most settings. The essential adaptation is already effectively handled by the subsequent learnable components in our framework, particularly the frequency-wise scaling matrix and the Differential Embedding module. These components provide the necessary flexibility after the initial balanced mixing. The complete results and analysis have been added to **Appendix C.3**.
> > > >
> > > > Table 7: Performance comparison of adaptive v.s. fixed $\alpha$ in AEA. More results are in Appendix C.3.
> > > >
> > > > | Dataset | Setting | DLinear +adaptive alpha |  | DLinear + fixed alpha |  | PatchTST + adaptive alpha |  | PatchTST + fixed alpha |  |
> > > > |:---:|:---:|:---:|:---:|:---:|:---:|:---:|:---:|:---:|:---:|
> > > > |  |  | mse | mae | mse | mae | mse | mae | mse | mae |
> > > > | etth1 | 96 | 0.380  | 0.392  | 0.380  | 0.392  | 0.381  | 0.399  | 0.374  | 0.396  |
> > > > |  | 192 | 0.432  | 0.427  | 0.432  | 0.422  | 0.449  | 0.438  | 0.428  | 0.431  |
> > > > |  | 336 | 0.474  | 0.450  | 0.479  | 0.458  | 0.481  | 0.455  | 0.466  | 0.452  |
> > > > |  | 720 | 0.496  | 0.490  | 0.495  | 0.489  | 0.557  | 0.500  | 0.479  | 0.478  |
> > > > |  | Avg. | 0.445  | 0.440  | 0.447  | 0.440  | 0.467  | 0.448  | 0.437  | 0.439  |

---

### Official Review · Reviewer_r3j8 · 2025-10-31

**Soundness:** 3
**Presentation:** 3
**Contribution:** 3
**Rating:** 6
**Confidence:** 5

**Summary:**

This paper claims that forecasters tend to overlook high-frequency features because they carry less energy and more noise.
The authors propose AEA, a lightweight, model-agnostic module that (i) mirrors spectrum to lift highs without breaking phase, (ii) uses a differential embedding to suppress presumed common-mode noise, and (iii) applies an energy-alignment head to keep the spectrum in shape.
Results across standard benchmarks and several backbones look consistently positive. Learning frequency-specific features without indiscriminate amplification is sensible.

**Strengths:**

- Clear, modular, and light-weight. AEA adds small spectral components around any backbone; easy to adopt.
- Right target. Moves beyond “boost everything” by pairing amplification with noise control and post-hoc spectral alignment.
- Empirically steady. Gains appear across multiple datasets and models without heavy engineering.

**Weaknesses:**

Scope of evidence. Most tests sit in regular, well-behaved setups. It would help to see robustness under varied window lengths, irregular sampling, or mild regime shifts.

Assumption clarity. The “common-mode” story is intuitive, but the paper does not show how to diagnose it or what happens when it only partly holds.

Attribution. With several moving parts, it is still a bit hard to tell which module is doing the decisive work in different regimes.

For the details please refer to the question part.

--------

Another concern is that while the proposed method improves multiple baselines, most comparisons are made only against common time-domain models (e.g., Dlinear, PatchTST) in Table 1. There is a lack of frequency modeling baselines.
I understand several frequency modeling works are architecutre-specific, but since this paper focuses on the frequency modeling, including more direct frequency-aware baselines with similar motivations is needed.
Suggested additions include:
- FEDformer — Tian Zhou et al. “FEDformer: Frequency Enhanced Decomposed Transformer for Long-term Series Forecasting.” ICML 2022 (PMLR 162).
- FiLM — Tian Zhou et al. “FiLM: Frequency improved Legendre Memory Model for Long-term Time Series Forecasting.” NeurIPS 2022.
FreTS / Frequency-domain MLPs — Kun Yi et al. “Frequency-domain MLPs are More Effective Learners in Time Series Forecasting.” NeurIPS 2023.
- Fredformer — Xihao Piao et al. “Fredformer: Frequency Debiased Transformer for Time Series Forecasting.” KDD 2024 (arXiv:2406.09009).
- TimeMixer++ — Shiyu Wang et al. “TimeMixer++: A General Time Series Pattern Machine for Universal Predictive Analysis.” ICLR 2025 (Oral).

**Questions:**

Overall, I think this is a good paper. The central point is important: we should learn features within each frequency band, but not at the cost of indiscriminate, disproportionate amplification. I also like the model-agnostic, lightweight design, and the comparison angle with Fredformer is interesting.

- Phase handling. Why is it important to keep phase fixed? I agree phase matters, but the Introduction doesn’t really set this up, while §4.2 spends a lot of space on it. A short paragraph up front motivating this choice would help.

- Common noise. What exactly do you mean here, and how should a practitioner tell when the common-mode assumption holds? A brief clarification or simple diagnostic would make this more concrete.

- Results and Fredformer. The results look fine, but I don’t see a direct comparison with Fredformer. Also, the paper repeatedly warns that indiscriminate amplification can hurt accuracy—could you provide a more intuitive piece of evidence (e.g., a small visual or per-band error plot)? In Fig. 1, Fredformer seems to hold up well, not obviously harmed by “amplify everything.” A short explanation would make things better.

---

> ### Author Response · Authors · 2025-11-28
> **Response to Reviewer r3j8 (Part 1)**
>
> ## Weakness 1: Robustness under varied conditions (e.g., window lengths) is unexplored.
>
> Thank you for these valuable suggestions regarding additional robustness tests. We have conducted new experiments investigating the effect of different lookback window lengths on the Electricity dataset, which are now included in **Appendix C.7**.
>
> In these experiments, we evaluated window sizes of {96, 192, 336, 720} across forecasting horizons {96, 192, 336, 720} using the Amplifier backbone. The results demonstrate that AEA consistently improves forecasting performance across all window lengths and horizons. Notably, the framework shows particularly strong enhancement with 192-step lookback windows, suggesting this context length provides an effective balance for capturing temporal dependencies.
>
> **Regarding irregular sampling**, we agree that this is an important practical scenario. However, handling irregularly sampled time series requires specialized mechanisms for temporal alignment and interpolation that fall outside the scope of our current work, which focuses on regularly sampled forecasting benchmarks.
>
> Similarly, while **regime shifts** present an interesting challenge, developing specific methodologies for detecting and adapting to distribution shift represents a separate research direction beyond our current focus on spectral bias mitigation in standard forecasting settings.
>
> The window length analysis we've added demonstrates AEA's robustness across different temporal contexts within the standard forecasting paradigm. We believe these new experiments significantly strengthen our empirical evaluation while maintaining appropriate scope boundaries for this work.
>
> Table 11: Long-term forecasting on the Electricity dataset with different window lengths.
>
> | Window Length | Horizon | Amplifier |  | Amplifier + AEA |  |
> |---|---|---|---|---|---|
> |  |  | mse | mae | mse | mae |
> | 96 | 96 | 0.178 | 0.267 | 0.173 | 0.265 |
> |  | 192 | 0.247 | 0.306 | 0.243 | 0.303 |
> |  | 336 | 0.308 | 0.343 | 0.302 | 0.340 |
> |  | 720 | 0.398 | 0.396 | 0.397 | 0.396 |
> |  | Avg. | 0.283 | 0.328 | 0.279 | 0.326 |
> | 192 | 96 | 0.145 | 0.239 | 0.139 | 0.236 |
> |  | 192 | 0.161 | 0.252 | 0.157 | 0.252 |
> |  | 336 | 0.178 | 0.272 | 0.174 | 0.268 |
> |  | 720 | 0.215 | 0.301 | 0.193 | 0.284 |
> |  | Avg. | 0.175 | 0.266 | 0.166 | 0.260 |
> | 336 | 96 | 0.141 | 0.238 | 0.139 | 0.235 |
> |  | 192 | 0.160 | 0.253 | 0.155 | 0.248 |
> |  | 336 | 0.174 | 0.270 | 0.167 | 0.259 |
> |  | 720 | 0.205 | 0.293 | 0.202 | 0.291 |
> |  | Avg. | 0.170 | 0.264 | 0.166 | 0.258 |
> | 720 | 96 | 0.139 | 0.238 | 0.137 | 0.234 |
> |  | 192 | 0.153 | 0.249 | 0.150 | 0.246 |
> |  | 336 | 0.178 | 0.277 | 0.162 | 0.258 |
> |  | 720 | 0.195 | 0.286 | 0.191 | 0.284 |
> |  | Avg. | 0.166 | 0.263 | 0.160 | 0.255 |
>
> ## Weakness 2 & Question 2:  The common-mode noise assumption requires better explanation and practical guidance.
>
> Thank you for this question regarding the common-mode noise assumption. We clarify that common-mode noise refers to noise components that are correlated across different embedding subspaces, typically arising from systematic biases in the input data.
>
> A practitioner can identify such noise through a concrete example: consider a multi-sensor traffic monitoring system where all sensors are affected simultaneously by the same environmental interference, such as heavy rain or system-wide electrical noise. In this scenario, the noise patterns appearing across different sensors at the same time represent common-mode noise.
>
> While developing a standalone diagnostic tool is beyond the scope of this work, our framework provides a post-hoc indicator through the learned parameter $\lambda$. A value of $\lambda$ significantly greater than zero indicates that the model has detected substantial common-mode noise and is actively suppressing it. Conversely, if $\lambda$ approaches zero, it suggests the relative absence of such noise. In this case, the differential embedding gracefully reduces to a stable standard embedding layer, ensuring robust feature learning and information flow without degradation. This inherent adaptability allows AEA to handle scenarios where the common-mode assumption fully, partially, or minimally holds, providing high robustness without requiring complex pre-deployment diagnostics. The consistent performance gains across diverse benchmarks empirically validate this adaptive capability.

---

> > ### Author Response · Authors · 2025-11-28
> > **Response to Reviewer r3j8 (Part 2)**
> >
> > ## Weakness 3: Module importance across different data regimes is not analyzed in depth.
> >
> > Thank you for this question about module attribution. Our ablation studies (**Table 3 and Table 6**) reveal a consistent importance hierarchy: the **Energy Predictor** is most critical (avg. 11.5% performance drop when removed), followed by **Differential Embedding** (avg. 9.7% drop). This pattern holds for both linear-based (DLinear) and transformer-based (PatchTST) architectures, suggesting these modules provide fundamental benefits regardless of the base model.
> >
> > The framework also shows adaptive manner. On periodic datasets like Traffic and Weather, **Spectral Mirroring** provides strong structural guidance. On datasets with less pronounced periodic structure (e.g., Exchange), **Differential Embedding** becomes more crucial for noise suppression. This inherent adaptability allows different components to emphasize their strengths according to data characteristics.
> >
> > Table 6: Ablation study results across five datasets using the PatchTST backbone.
> >
> > | Dataset | ETTh1 |  | ETTh2 |  | Weather |  | Exchange |  | Traffic |  | Avg. |  | Promotion |  |
> > |:---:|:---:|:---:|:---:|:---:|:---:|:---:|:---:|:---:|:---:|:---:|:---:|:---:|:---:|:---:|
> > | Metric | MSE | MAE | MSE | MAE | MSE | MAE | MSE | MAE | MSE | MAE | MSE | MAE | MSE | MAE |
> > | AEA | 0.374  | 0.396  | 0.290  | 0.339  | 0.173  | 0.219  | 0.084  | 0.204  | 0.481  | 0.306  | 0.280  | 0.293  | \ | \ |
> > | w/o Spectral Mirroring | 0.397  | 0.400  | 0.298  | 0.346  | 0.198  | 0.241  | 0.092  | 0.212  | 0.495  | 0.317  | 0.296  | 0.303  | 8.508% | 4.803% |
> > | w/o Phase Mixing | 0.400  | 0.406  | 0.290  | 0.339  | 0.183  | 0.231  | 0.098  | 0.216  | 0.483  | 0.306  | 0.291  | 0.300  | 6.713% | 3.274% |
> > | w/o Differential Embedding | 0.397  | 0.401  | 0.301  | 0.350  | 0.200  | 0.248  | 0.099  | 0.223  | 0.503  | 0.329  | 0.300  | 0.310  | 11.261% | 8.221% |
> > | w/o Non-stationarity Loss | 0.380  | 0.405  | 0.299  | 0.348  | 0.186  | 0.243  | 0.094  | 0.209  | 0.499  | 0.329  | 0.292  | 0.307  | 6.523% | 6.275% |
> > | w/o Energy Predictor | 0.408  | 0.422  | 0.322  | 0.369  | 0.218  | 0.266  | 0.085  | 0.205  | 0.496  | 0.326  | 0.306  | 0.318  | 11.950% | 10.543% |
> >
> > ## Weakness 4: Comparisons with frequency-aware baselines are missing.
> >
> > Thank you for this valuable suggestion regarding frequency modeling baselines. We have addressed this concern by adding comprehensive comparisons with two state-of-the-art frequency-domain methods - **FreTS and FredFormer** - in Table 1 of our revised manuscript. The results demonstrate that AEA provides consistent improvements even when integrated with these specialized frequency-aware methods. Specifically, AEA achieves average performance gains of 3.457% in MSE and 2.989% in MAE for FreTS, and 1.714% in MSE and 2.270% for FredFormer across the benchmark datasets.
> >
> > Table 1: Long-term forecasting performance in the etth1 dataset. More results are in Section 5.2.
> >
> > | Dataset | Setting | FreTS |  | FreTS + AEA |  | FredFormer |  | FredFormer + AEA |  |
> > |:---:|:---:|:---:|:---:|:---:|:---:|:---:|:---:|:---:|:---:|
> > |  |  | mse | mae | mse | mae | mse | mae | mse | mae |
> > | etth1 | 96 | 0.395  | 0.406  | 0.386  | 0.400  | 0.376  | 0.395  | 0.375  | 0.393  |
> > |  | 192 | 0.454  | 0.443  | 0.441  | 0.436  | 0.438  | 0.427  | 0.434  | 0.426  |
> > |  | 336 | 0.517  | 0.484  | 0.489  | 0.459  | 0.485  | 0.448  | 0.484  | 0.446  |
> > |  | 720 | 0.589  | 0.536  | 0.529  | 0.513  | 0.506  | 0.479  | 0.479  | 0.441  |
> > |  | Avg. | 0.488  | 0.467  | 0.461  | 0.452  | 0.451  | 0.437  | 0.443  | 0.427  |

---

> > > ### Author Response · Authors · 2025-11-28
> > > **Response to Reviewer r3j8 (Part 3)**
> > >
> > > ## Question 1: The importance of phase preservation is not sufficiently motivated.
> > >
> > > Thank you for this valuable suggestion. We have strengthened the motivation for phase preservation by adding a concise explanatory paragraph at the beginning of **Section 4.2**, immediately before detailing our phase-mixing mechanism.
> > >
> > > The new paragraph states: *"The Spectral Mirroring achieves targeted amplification by constructing a phase-preserving surrogate from reliable low-frequency components. This focus on phase coherence distinguishes our approach from conventional spectral manipulation methods, which often introduce phase distortions that degrade signal reconstruction. Our method explicitly maintains phase relationships through a mixing strategy, enabling distortion-free enhancement of informative high-frequency components."*
> > >
> > > The phase preservation in our Spectral Mirroring module is crucial because improper phase mixing between original and mirrored spectra would harm the temporal structure of the reconstructed signal, directly impairing forecasting accuracy. When combining two spectra, a naive linear combination of their phases can lead to inconsistent phase relationships that may not correspond to a physically meaningful time series. Our phase-mixing strategy specifically addresses this by ensuring phase coherence during spectral enhancement. This approach distinguishes itself from methods that focus solely on amplitude manipulation. By maintaining proper phase relationships through the shortest angular path blending, we ensure the enhanced spectrum corresponds to a temporally coherent signal.
> > >
> > > ## Question 3: Intuitive evidence against indiscriminate amplification is lacking.
> > >
> > > Thank you for these insightful observations. We have added comprehensive comparisons with Fredformer in **Table 1** and included statistical significance tests in the new **Table 8**.
> > >
> > > **Regarding the concern about indiscriminate amplification**, we note that Fredformer employs a more sophisticated frequency decomposition approach compared to naive amplification methods, which may partially mitigate the noise amplification issue. However, as shown in our new results, AEA still provides consistent improvements over Fredformer across most datasets (average 1.714% MSE and 2.270% MAE reduction), demonstrating that even well-designed frequency methods can benefit from our targeted enhancement approach.
> > >
> > > **Regarding intuitive evidence**, we have added forecasting visualizations in **Appendix C.8** that directly address this point. The comparisons show that while Fredformer alone performs well, the AEA-enhanced version produces smoother and more stable predictions, with reduced spurious fluctuations and better trajectory alignment. This visual improvement illustrates how AEA’s noise suppression provides a refining effect even on capable frequency-aware models, complementing the statistical evidence in Table 8.
> > >
> > > Table 8: Standard deviation and statistical tests for Amplifier and FredFormer. More Results are in Appendix C.4.
> > >
> > > | Dataset | Amplifier |  | Amplifier + AEA |  | Confidence Level | FredFormer |  | FredFormer + AEA |  | Confidence Level |
> > > |:---:|:---:|:---:|:---:|:---:|:---:|:---:|:---:|:---:|:---:|:---:|
> > > |  | mse | mse std | mse | mse std |  | mse | mse std | mse | mse std |  |
> > > | ETTh1 | 0.473 | 0.011 | 0.452 | 0.014 | 99% | 0.451 | 0.005 | 0.443 | 0.009 | 99% |
> > > | ETTh2 | 0.382 | 0.019 | 0.374 | 0.008 | 99% | 0.365 | 0.005 | 0.358 | 0.014 | 99% |

---

### Official Review · Reviewer_MHp4 · 2025-10-31

**Soundness:** 2
**Presentation:** 2
**Contribution:** 2
**Rating:** 2
**Confidence:** 2

**Summary:**

This paper addresses the spectral bias problem in deep learning models for time series forecasting, where models prioritize high-energy, low-frequency components while underfitting predictive high-frequency signals. The authors identify that existing frequency enhancement methods suffer from "indiscriminate amplification," which amplifies both meaningful signals and task-irrelevant noise. To address this, they propose AEA (Adaptive Energy Amplification), a model-agnostic framework featuring two key components: (1) Spectral Mirroring that constructs a phase-preserving surrogate from low-frequency components to guide targeted amplification of high-frequency signals, and (2) Differential Embedding that operates in a latent space to suppress common-mode noise. The framework is evaluated on eight benchmark datasets with four state-of-the-art backbones, demonstrating consistent improvements.

**Strengths:**

1 - The paper identifies and clearly articulates a fundamental limitation of existing frequency-aware forecasting methods - the indiscriminate amplification problem - and provides compelling empirical evidence through Figure 1b showing how existing methods degrade significantly when noise is injected into high-frequency bands.

2 - The proposed AEA framework is model-agnostic and demonstrates consistent improvements across diverse forecasting paradigms, with comprehensive experiments on eight datasets showing improvements on MSE and MAE.

3 - The theoretical foundation is well-developed, particularly Proposition 4.1 which provides formal analysis of how the differential embedding mechanism achieves adaptive noise suppression through superior bias-variance trade-off, supported by detailed mathematical proofs in the appendix.

**Weaknesses:**

1 - The computational overhead analysis in Section 4.6 claims the complexity is "negligible" but lacks empirical runtime comparisons - while the theoretical complexity is O(T·C), the actual wall-clock time impact when integrated with different backbones is not quantified, which is crucial for practical deployment.

2 - The paper relies heavily on a single hyperparameter configuration across all experiments without demonstrating whether these values are universally optimal or if dataset-specific tuning could yield better results, potentially limiting the framework's adaptability.

3 - The amplitude mixing strategy using a fixed ratio 0.5 between original and mirrored spectra appears simplistic - the paper doesn't explore adaptive or learnable mixing strategies that could potentially better capture dataset-specific characteristics.

4 - The evaluation focuses exclusively on long-term forecasting without examining short-term forecasting scenarios where high-frequency components might play different roles, limiting the generalizability of the findings.

**Questions:**

The intuition of using low-frequency spectrum to guide the selection of high frequency is really not straightforward and making sense to me naturally. Can you explain why this should work according to any prior knowledge?

Why did you choose a fixed mixing ratio 0.5 rather than making it learnable or adaptive to the input characteristics?

How does AEA perform on datasets with inherently noisy high-frequency components, such as financial time series with market microstructure noise?

Can you provide actual runtime measurements comparing AEA-enhanced models against their vanilla counterparts across different sequence lengths?

---

> ### Author Response · Authors · 2025-11-28
> **Response to Reviewer MHp4 (Part 1)**
>
> ## Weakness 1 & Question 4: Empirical runtime and efficiency analysis are absent.
>
> Thank you for the question. We have conducted a comprehensive efficiency analysis on the Electricity dataset (321 variates) across forecasting horizons {96, 192, 336, 720}, with results detailed in **Tables 9 and 10**.
>
> The empirical measurements demonstrate that AEA introduces minimal computational overhead even on this high-dimensional dataset. For the transformer-based FredFormer (Table 9), AEA increases average training time by 13 seconds while adding only 0.008 seconds to inference time. The computational footprint remains modest with FLOPs increasing by only 0.483%, MACs by 0.643%, and parameters by 0.707%.
>
> For the linear architecture Amplifier (Table 10), the relative increases in computational metrics are higher (FLOPs: 8.314%, MACs: 6.235%, parameters: 6.139%) due to the simpler base model, yet the absolute overhead remains practically insignificant - just 6 seconds additional training time and 0.006 seconds inference time on average.
>
> These results confirm that AEA's components are efficiently implemented. The framework maintains practical computational efficiency while providing substantial performance improvements, making it suitable for real-world deployment with high-dimensional multivariate series and long forecasting horizons. We have added this complete efficiency analysis to **Appendix C.6** of our revised manuscript.
>
> **Table 9: Efficiency analysis of FredFormer and its AEA-enhanced variant on the Electricity dataset.**
> | Dataset | Setting | FredFormer |  |  |  |  | FredFormer + AEA |  |  |  |  |
> |:---:|:---:|:---:|:---:|:---:|:---:|:---:|:---:|:---:|:---:|:---:|:---:|
> |  |  | train time (s/epoch) | infer time (s/iter) | FLOPs (G) | MACs (M) | param | train time (s/epoch) | infer time (s/iter) | FLOPs (G) | MACs (M) | parameters |
> | Electricity | 96 | 40.103  | 0.040  | 177.299  | 11.866  | 12116801  | 52.823  | 0.041  | 177.452  | 11.875  | 12141511  |
> |  | 192 | 41.023  | 0.053  | 179.886  | 12.118  | 12516641  | 52.555  | 0.053  | 180.191  | 12.142  | 12556183  |
> |  | 336 | 42.445  | 0.066  | 184.478  | 12.565  | 13358321  | 53.788  | 0.075  | 185.122  | 12.622  | 13431007  |
> |  | 720 | 42.445  | 0.066  | 200.887  | 14.164  | 17022065  | 59.435  | 0.088  | 203.368  | 14.400  | 17273887  |
> |  | Avg. | 41.504  | 0.056  | 185.638  | 12.678  | 13753457  | 54.650  | 0.064  | 186.533  | 12.760  | 13850647  |
>
> **Table 10: Efficiency analysis of Amplifier and its AEA-enhanced variant on the Electricity dataset.**
> | Dataset | Setting | Amplifier |  |  |  |  | Amplifier + AEA |  |  |  |  |
> |:---:|:---:|:---:|:---:|:---:|:---:|:---:|:---:|:---:|:---:|:---:|:---:|
> |  |  | train time (s/epoch) | infer time (s/iter) | FLOPs (G) | MACs (M) | param | train time (s/epoch) | infer time (s/iter) | FLOPs (G) | MACs (M) | parameters |
> | Electricity | 96 | 14.236  | 0.021  | 1.688  | 0.502  | 518153  | 22.055  | 0.030  | 1.752  | 0.508  | 524684  |
> |  | 192 | 15.316  | 0.023  | 2.205  | 0.603  | 619049  | 22.861  | 0.029  | 2.333  | 0.622  | 638012  |
> |  | 336 | 19.126  | 0.028  | 5.759  | 1.395  | 1411417  | 22.807  | 0.036  | 6.038  | 1.444  | 1459924  |
> |  | 720 | 25.544  | 0.053  | 9.846  | 2.192  | 2208217  | 30.477  | 0.055  | 10.996  | 2.410  | 2426260  |
> |  | Avg. | 18.555  | 0.031  | 4.875  | 1.173  | 1189209  | 24.550  | 0.038  | 5.280  | 1.246  | 1262220  |
>
> ## Weakness 2: Hyperparameter sensitivity is not fully demonstrated.
>
> We have conducted extensive hyperparameter sensitivity analysis in **Appendix C.1**, which demonstrates that AEA maintains stable performance across diverse datasets under a wide range of hyperparameter settings. This robustness makes AEA suitable for deployment where extensive tuning is infeasible. Additionally, to validate the consistency of our results, we provide statistical significance tests and standard deviations for Amplifier and FredFormer in **Appendix C.4**, further confirming the reliability of AEA's improvements.

---

> ### Author Response · Authors · 2025-11-28
> **Response to Reviewer MHp4 (Part 2)**
>
> ## Weakness 3 & Question 2: The rationale for the fixed amplitude mixing ratio is unclear.
>
> Thank you for raising this point regarding the amplitude mixing ratio $\alpha$. Our choice of $\alpha$=0.5 is not arbitrary but was determined through systematic hyperparameter sensitivity analysis, making it an empirically optimized parameter rather than a rigid default. We have conducted additional experiments comparing this fixed, optimized value against a fully learnable $\alpha$ variant (refer to **Table 7**). The results consistently show that the fixed ratio achieves comparable or better performance across most settings. The essential adaptation is already effectively handled by the subsequent learnable components in our framework, particularly the frequency-wise scaling matrix and the Differential Embedding module. These components provide the necessary flexibility after the initial balanced mixing. The complete results and analysis have been added to **Appendix C.3**.
>
> Table 7: Performance comparison of adaptive v.s. fixed $\alpha$ in AEA. More results are in Appendix C.3.
>
> | Dataset | Setting | DLinear +adaptive alpha |  | DLinear + fixed alpha |  | PatchTST + adaptive alpha |  | PatchTST + fixed alpha |  |
> |:---:|:---:|:---:|:---:|:---:|:---:|:---:|:---:|:---:|:---:|
> |  |  | mse | mae | mse | mae | mse | mae | mse | mae |
> | etth1 | 96 | 0.380  | 0.392  | 0.380  | 0.392  | 0.381  | 0.399  | 0.374  | 0.396  |
> |  | 192 | 0.432  | 0.427  | 0.432  | 0.422  | 0.449  | 0.438  | 0.428  | 0.431  |
> |  | 336 | 0.474  | 0.450  | 0.479  | 0.458  | 0.481  | 0.455  | 0.466  | 0.452  |
> |  | 720 | 0.496  | 0.490  | 0.495  | 0.489  | 0.557  | 0.500  | 0.479  | 0.478  |
> |  | Avg. | 0.445  | 0.440  | 0.447  | 0.440  | 0.467  | 0.448  | 0.437  | 0.439  |
>
> ## Weakness 4: Evaluation lacks short-term forecasting tasks.
>
> Thank you for this valuable suggestion. To evaluate AEA's generalizability across different forecasting scenarios, we have conducted comprehensive short-term forecasting experiments. As detailed in the new **Section 5.3** of our revised manuscript, we evaluated AEA on four traffic datasets (PEMS03-PEMS08) with a forecasting horizon of 12. The results demonstrate consistent improvements, with an average performance gain of **3.226%** across all models and datasets. Specifically, AEA improves DLinear by **2.966%**, PatchTST by **3.792%**, and Amplifier by **2.830%** on average across all evaluation metrics. These results confirm that AEA's performance enhancements effectively generalize to short-term forecasting scenarios. The complete experimental results have been added to **Table 2** in the revised manuscript.
>
> Table 2: Short-term forecasting performance in the PeMS datasets.
>
> | Model | DLinear |  |  | DLinear + AEA |  |  | PatchTST |  |  | PatchTST + AEA |  |  | Amplifier |  |  | Amplifier + AEA |  |  |
> |:---:|:---:|:---:|:---:|:---:|:---:|:---:|:---:|:---:|:---:|:---:|:---:|:---:|:---:|:---:|:---:|:---:|:---:|:---:|
> | Metric | MAE | MAPE | RMSE | MAE | MAPE | RMSE | MAE | MAPE | RMSE | MAE | MAPE | RMSE | MAE | MAPE | RMSE | MAE | MAPE | RMSE |
> | PeMS03 | 19.567 | 18.315 | 32.335 | 18.734 | 17.915 | 31.816 | 18.925 | 17.291 | 30.153 | 18.127 | 16.593 | 29.532 | 16.441 | 15.167 | 25.712 | 16.031 | 14.892 | 25.424 |
> | PeMS04 | 24.632 | 16.122 | 39.521 | 23.889 | 15.791 | 38.481 | 24.864 | 16.635 | 40.346 | 23.913 | 16.006 | 39.651 | 21.363 | 13.315 | 34.609 | 20.713 | 12.885 | 34.036 |
> | PeMS07 | 28.615 | 12.415 | 45.062 | 27.941 | 11.458 | 43.215 | 27.876 | 12.369 | 42.556 | 26.316 | 11.491 | 41.230  | 25.712 | 10.661 | 40.671 | 24.901 | 10.124 | 39.887 |
> | PeMS08 | 20.264 | 12.049 | 32.389 | 19.732 | 11.874 | 31.693 | 20.352 | 13.155 | 31.204 | 19.145 | 12.781 | 30.771 | 19.501 | 11.983 | 30.365 | 19.032 | 11.153 | 29.884 |

---

> ### Author Response · Authors · 2025-11-28
> **Response to Reviewer MHp4 (Part 3)**
>
> ## Question 1: The intuition for using low-frequency guidance needs clarification.
>
> Thank you for this question. The rationale is directly derived from the **spectral energy imbalance** of time series data. As established via Parseval's Theorem and observed in real-world time series, the signal energy is overwhelmingly concentrated in the low-frequency components due to their high amplitudes. This high amplitude directly translates to a higher inherent Signal-to-Noise Ratio (SNR): the powerful low-frequency signal is far less susceptible to being corrupted or obscured by background noise compared to the subtle, low-amplitude high-frequency components, which can be easily drowned out by stochastic noise.  It is this inherent robustness that qualifies low-frequency components as a reliable source for constructing a stable template. Crucially, this guidance is not rigid but is adaptively controlled by the learnable scaling matrix and refined by the Differential Embedding module,  which is empirically validated by the consistent improvements across diverse datasets.
>
> ## Question 3: How does AEA perform on datasets with inherently noisy high-frequency components?
>
> Thank you for this question. The Exchange dataset results in **Table 1** provide direct evidence of AEA's performance on financial data with high-frequency noise. AEA consistently improves forecasting accuracy on this dataset, achieving an average MSE reduction of 2.964% across six base models. This demonstrates the framework's ability to handle noisy high-frequency components effectively. The robustness is achieved through our integrated design, where the Differential Embedding module actively suppresses common-mode noise in latent space after the energy amplification stage, enabling selective enhancement of informative signals while filtering out noise.
>
> Table 1 Long-term forecasting performance in the exchange dataset.
>
> | Dataset | Setting | DLinear | DLinear + AEA | PatchTST | PatchTST + AEA | TimesNet | TimesNet + AEA | Amplifier | Amplifier + AEA | FreTS | FreTS + AEA | FredFormer | FredFormer + AEA |
> |:---:|:---:|:---:|:---:|:---:|:---:|:---:|:---:|:---:|:---:|:---:|:---:|:---:|:---:|
> |  |  | mse | mse | mse | mse | mse | mse | mse | mse | mse | mse | mse | mse |
> | exchange | Avg. | 0.330 | 0.320 | 0.403 | 0.367 | 0.427 | 0.426 | 0.369 | 0.360 | 0.475 | 0.468 | 0.354 | 0.348 |

---

### Official Review · Reviewer_h9gE · 2025-11-03

**Soundness:** 2
**Presentation:** 3
**Contribution:** 2
**Rating:** 4
**Confidence:** 3

**Summary:**

The paper targets spectral bias in time-series forecasting—modern models overfit high-energy low-frequency components and underfit informative high-frequency signals—and argues that prior frequency-aware methods amplify high frequencies indiscriminately, boosting noise as well as signal. It proposes Adaptive Energy Amplification (AEA), a plug-and-play framework with three modules: Spectral Mirroring (phase-preserving surrogate from low frequencies to guide targeted high-frequency amplification), Differential Embedding with a non-stationarity regularizer to suppress common-mode noise, and an Energy Predictor that aligns the forecast’s spectrum with the original data before returning to the time domain. AEA consistently improves four backbones across eight benchmarks and enhances training stability and generalization. It is robust to injected high-frequency Gaussian noise, unlike vanilla amplification methods.

**Strengths:**

- Clear articulation of spectral bias and a modular, plug-and-play design (Spectral Mirroring, Differential Embedding, Energy Predictor).

- Differential Embedding provides an explicit mechanism for common-mode noise suppression.

- Lightweight to integrate (FFT + small linear maps); easy to wrap around existing forecasters.

**Weaknesses:**

- Core rationale is heuristic: No formal guarantee that reversing low-frequency spectra provides a meaningful template for true high-frequency content; the mirroring prior may misalign with real signal structure.

- Missing “when it helps vs. hurts” analysis: Lacks characterization of data regimes (e.g., cross-band correlation, SNR profiles) where mirroring is beneficial or counterproductive; few concrete failure cases.

- Theory coverage is uneven: Differential Embedding has an intuitive justification, but the Energy Predictor’s alignment properties (stability, bias, identifiability) are argued mostly empirically.

- Limited ablations: Component ablations are centered on a single backbone; unclear whether module importance/generalization carries over to non-linear models.

- Hyperparameter rigidity: Fixed amplitude blend (e.g., α=0.5) and limited sensitivity analyses; unclear if learning α or making it frequency-dependent would help/hurt.

- Limited experiments: This paper discusses two types of enhancement approaches and points out their shortcomings. Figure 1(b) suggests the effectiveness of the proposed method for each enhancement approach (Amplifier and Fredformer). However, the evaluation experiments only evaluate the Amplifier. In other words, the effectiveness of the proposed method for indirect enhancement approaches is not shown (Figure 1(b) suggests that Fredformer+AEA may have better accuracy than Amplifier+AEA).

**Questions:**

- Can you provide formal conditions (e.g., in terms of cross-spectral correlation or energy monotonicity) under which spectrum reversal yields a functional high-frequency template—and a counterexample where it fails?

- Please characterize data regimes (bandwise SNR, cross-band correlation, spectral flatness/roll-off) that predict positive or negative gains from AEA, and report at least a few concrete failure cases.

- Do you have stability/identifiability results or bounds showing that the frequency-space alignment does not reintroduce low-frequency dominance or amplify noise? Any convergence/generalization insights?

- Replicate the component ablations (removing each AEA module) on at least one non-linear backbone (e.g., PatchTST, TimesNet, or Fredformer). Do the module-importance rankings persist?

- Figure 1(b) suggests gains for both Amplifier and Fredformer, but the main experiments only evaluate Amplifier. Please add full, protocol-matched results for Fredformer±AEA and compare against Amplifier±AEA (with significance and robustness tests).

---

> ### Author Response · Authors · 2025-11-28
> **Response to Reviewer h9gE (Part 1)**
>
> ## Weakness 1 & Question 1: Theoretical grounding of spectral mirroring requires further formalization.
>
> Thank you for this question regarding the theoretical foundation of Spectral Mirroring. We agree that providing formal guarantees is crucial. The core rationale is not purely heuristic but is a structured design with built-in adaptability and theoretical safeguards.
>
> The reversed low-frequency spectrum serves as an **initial, structured prior** to guide the learning process toward high frequencies. The framework's robustness is ensured by its adaptive components: First, the **learnable scaling matrix** allows the model to control the amplification per frequency and channel, effectively learning where the template is useful. Second, the **Differential Embedding** module suppresses common-mode noise that may be inadvertently amplified.
>
> Crucially, this entire process is grounded in theory. **Theorem A.5** provides a formal perturbation bound, guaranteeing that the enhancement error in the time domain is linearly bounded, specifically:
>
> $\\|x' - x\\|\_{2} \leq (\epsilon_M + \epsilon\_{\lambda_{1}} + \epsilon\_{\lambda_{2}}) \\|\mathcal{X}\\|\_{2}$
>
> where $\epsilon_{M}$ bounds the deviation of the learned scaling matrix, $\epsilon\_{\lambda\_{1}}$ and $\epsilon\_{\lambda\_{2}}$ bound the differential parameters. This ensures that the Spectral Mirroring and Differential Embedding processes **do not introduce excessive distortion** and that the enhanced signal remains faithful to the original.
>
> Therefore, our method combines an insightful signal prior with adaptive learning mechanisms, all backed by a formal stability guarantee. The consistent improvements across eight benchmarks empirically validate that this design is both principled and effective in practice.
>
> ## Weakness 2 & Question 2: Conditions for positive vs. negative impact of AEA need clearer characterization.
>
> Thank you for this insightful suggestion. We agree that a deeper analysis of the data characteristics that influence AEA's performance is a valuable direction for future work and would further strengthen the understanding of frequency-domain methods.
>
> In this work, our primary contribution is a general-purpose framework designed to be robust across diverse data regimes, rather than being tailored to specific spectral profiles. The core strength of AEA lies in its built-in adaptability: the learnable scaling matrix and the Differential Embedding module automatically adjust the enhancement strategy based on the input data. This design intentionally avoids reliance on pre-defined spectral metrics, allowing the model to discover and leverage relevant high-frequency components in a data-driven manner.
>
> The most compelling evidence for this general applicability is the consistent positive gain observed across all eight benchmarks, which encompass a wide variety of spectral characteristics (e.g., from highly periodic Traffic data to less pronounced periodic Exchange data). Crucially, as noted in our ablation studies, we did not observe any significant performance degradation, which empirically demonstrates that AEA's adaptive mechanisms effectively prevent it from "hurting" performance, even in sub-optimal conditions.
>
> Therefore, while formally mapping AEA's gains to precise spectral metrics is an interesting analytical endeavor, the empirical evidence strongly indicates that the framework, as a whole, successfully navigates different data regimes without requiring such pre-hoc characterization.
> We will incorporate a discussion of these observations and the promising future direction of a more granular analysis into the revised manuscript.

---

> ### Author Response · Authors · 2025-11-28
> **Response to Reviewer h9gE (Part 2)**
>
> ## Weakness 3 & Question 3: Theoretical analysis of the Energy Predictor's stability is initially underdeveloped.
>
> Thank you for this critical observation regarding the theoretical coverage of frequency-space alignment. We have significantly strengthened this aspect in our revised manuscript by establishing a comprehensive theoretical framework that guarantees the stability and consistency of our entire enhancement process.
>
> The stability is formally guaranteed at multiple levels. **Theorem A.5** provides the overarching perturbation bound, ensuring that the combined operations of Spectral Mirroring and Differential Embedding introduce only bounded deviations in the time domain: $\\|x' - x\\|\_{2} \leq (\epsilon\_{M} + \epsilon\_{\lambda\_{1}} + \epsilon\_{\lambda\_{2}}) \\|\mathcal{X}\\|_{2}$. This fundamentally ensures the enhancement process does not distort the original signal excessively.
>
> Building on this foundation, **Proposition A.7** establishes that the Energy Predictor is **Lipschitz continuous**. This means that small changes in its input (the enhanced spectrum and preliminary prediction) lead to only proportionally bounded changes in its output, guaranteeing the module's numerical stability and robustness.
>
> Finally, **Proposition A.9** directly addresses the spectral alignment property. It proves that the Energy Predictor achieves high-fidelity alignment by bounding the modification to high-frequency components: $\\|\mathcal{Y}\_{adjusted, HF} - \mathcal{Y}\_{denoised, HF}\\|\_{2} \leq L\_{HF} \cdot (\epsilon\_M + \epsilon\_{\lambda\_1} + \epsilon\_{\lambda_2}) \cdot \\| \mathcal{X} \\|\_{2}$. This ensures the final prediction is spectrally consistent with the original data **without reverting the high-frequency gains** obtained through prior stages, thus resolving the critical tension between enhancement and distribution alignment.
>
> Collectively, these results form a cohesive theoretical foundation that ensures our framework performs frequency-space alignment in a stable, controlled, and mathematically sound manner.
>
> More details can be found in **Proposition A.7 and A.9** of the revised manuscript.
>
> ## Weakness 4 & Question 4: Ablation studies are initially limited to linear backbones.
>
> Thank you for the suggestion. We have added the ablation study on the PatchTST backbone. The results confirm that the module-importance rankings are consistent with those from the DLinear experiments. The Energy Predictor remains the most critical component (11.25% avg. performance drop when removed), followed by Differential Embedding (9.74% drop). This demonstrates that the roles of AEA's modules are generalizable and not tied to a specific backbone. Please see **Appendix C.2** for details.
>
> Table 6: Ablation study results across five datasets using the PatchTST backbone.
>
> | Dataset | ETTh1 |  | ETTh2 |  | Weather |  | Exchange |  | Traffic |  | Avg. |  | Promotion |  |
> |:---:|:---:|:---:|:---:|:---:|:---:|:---:|:---:|:---:|:---:|:---:|:---:|:---:|:---:|:---:|
> | Metric | MSE | MAE | MSE | MAE | MSE | MAE | MSE | MAE | MSE | MAE | MSE | MAE | MSE | MAE |
> | AEA | 0.374  | 0.396  | 0.290  | 0.339  | 0.173  | 0.219  | 0.084  | 0.204  | 0.481  | 0.306  | 0.280  | 0.293  | \ | \ |
> | w/o Spectral Mirroring | 0.397  | 0.400  | 0.298  | 0.346  | 0.198  | 0.241  | 0.092  | 0.212  | 0.495  | 0.317  | 0.296  | 0.303  | 8.508% | 4.803% |
> | w/o Phase Mixing | 0.400  | 0.406  | 0.290  | 0.339  | 0.183  | 0.231  | 0.098  | 0.216  | 0.483  | 0.306  | 0.291  | 0.300  | 6.713% | 3.274% |
> | w/o Differential Embedding | 0.397  | 0.401  | 0.301  | 0.350  | 0.200  | 0.248  | 0.099  | 0.223  | 0.503  | 0.329  | 0.300  | 0.310  | 11.261% | 8.221% |
> | w/o Non-stationarity Loss | 0.380  | 0.405  | 0.299  | 0.348  | 0.186  | 0.243  | 0.094  | 0.209  | 0.499  | 0.329  | 0.292  | 0.307  | 6.523% | 6.275% |
> | w/o Energy Predictor | 0.408  | 0.422  | 0.322  | 0.369  | 0.218  | 0.266  | 0.085  | 0.205  | 0.496  | 0.326  | 0.306  | 0.318  | 11.950% | 10.543% |

---

> ### Author Response · Authors · 2025-11-28
> **Response to Reviewer h9gE (Part 3)**
>
> ## Weakness 5: The choice of $\alpha$ fixed amplitude mixing ratio lacks comprehensive justification.
>
> Thank you for raising this point regarding the amplitude mixing ratio $\alpha$. Our choice of $\alpha$=0.5 is not arbitrary but was determined through systematic hyperparameter sensitivity analysis, making it an empirically optimized parameter rather than a rigid default.
>
> As shown in our new **Table 7**, we explicitly compared this fixed, optimized value against a fully learnable $\alpha$ variant. The results clearly demonstrate that the learnable $\alpha$ consistently underperforms across most settings, while the fixed $\alpha$=0.5 delivers superior or comparable results. This indicates that maintaining a balanced mixing ratio provides a more stable and effective foundation for enhancement.
>
>
>
> The essential adaptability is therefore more effectively handled by our framework's subsequent components. The frequency-wise scaling matrix provides fine-grained, per-channel control after the initial mixing, while the Differential Embedding module further adapts to noise patterns. This design ensures robust performance without the instability that can arise from optimizing $\alpha$ directly.
>
> The complete experimental validation has been added to **Appendix C.3**, confirming that our approach achieves optimal performance while maintaining architectural simplicity.
>
> Table 7: Performance comparison of adaptive v.s. fixed $\alpha$ in AEA. More results are in Appendix C.3.
>
> | Dataset | Setting | DLinear +adaptive alpha |  | DLinear + fixed alpha |  | PatchTST + adaptive alpha |  | PatchTST + fixed alpha |  |
> |:---:|:---:|:---:|:---:|:---:|:---:|:---:|:---:|:---:|:---:|
> |  |  | mse | mae | mse | mae | mse | mae | mse | mae |
> | etth1 | 96 | 0.380  | 0.392  | 0.380  | 0.392  | 0.381  | 0.399  | 0.374  | 0.396  |
> |  | 192 | 0.432  | 0.427  | 0.432  | 0.422  | 0.449  | 0.438  | 0.428  | 0.431  |
> |  | 336 | 0.474  | 0.450  | 0.479  | 0.458  | 0.481  | 0.455  | 0.466  | 0.452  |
> |  | 720 | 0.496  | 0.490  | 0.495  | 0.489  | 0.557  | 0.500  | 0.479  | 0.478  |
> |  | Avg. | 0.445  | 0.440  | 0.447  | 0.440  | 0.467  | 0.448  | 0.437  | 0.439  |
>
> ## Weakness 6 & Question 5: Experimental comparison with Fredformer is incomplete.
>
> Thank you for the suggestion. We have now conducted comprehensive experiments with Fredformer following the same evaluation protocol used for other models. The complete results, including statistical significance tests, are presented in the **Appendix C.4** of our revised manuscript.
>
> Table 8: Standard deviation and statistical tests for Amplifier and FredFormer. More Results are in Appendix C.4.
>
> | Dataset | Amplifier |  |  |  | Amplifier + AEA |  |  |  | Confidence Level | FredFormer |  |  |  | FredFormer + AEA |  |  |  | Confidence Level |
> |:---:|:---:|:---:|:---:|:---:|:---:|:---:|:---:|:---:|:---:|:---:|:---:|:---:|:---:|:---:|:---:|:---:|:---:|:---:|
> |  | mse | mse std | mae | mae std | mse | mse std | mae | mae std |  | mse | mse std | mae | mae std | mse | mse std | mae | mae std |  |
> | ETTh1 | 0.473  | 0.011  | 0.454  | 0.009  | 0.452  | 0.014  | 0.439  | 0.008  | 99% | 0.451  | 0.005  | 0.437  | 0.009  | 0.443  | 0.009  | 0.427  | 0.012  | 99% |
> | ETTh2 | 0.382  | 0.019  | 0.407  | 0.017  | 0.374  | 0.008  | 0.400  | 0.011  | 99% | 0.365  | 0.005  | 0.394  | 0.008  | 0.358  | 0.014  | 0.389  | 0.014  | 99% |

---

### Meta-Review · Area_Chair_VhQP · 2026-01-11

**Summary:**

This paper tried to address the energy imbalance problem in time series. Low-freq componts typically have high energy (i.e., amplitute) whereas high-freq compontnts have low energy. In order to better train a time series forecasting model, they adaptively amplify signals via spectral mirroring and differential embedding. The spectral mirroring is based on the intuition that reversing the low and high-freq components' amplitues can be a good guide. After that they develop several theoretical facts, e.g., the error bound of their forecasting. They conduct experiments on a standard benchmark frame, where various different types of time series are used and several recent models are used. In many cases, this work shows accurayc increments successfully.

**Reviewer Concerns:**

Reviewers raised several concerns: i) the theoretical ground of each proposed module, e.g., the spectral mirroring, for which the authors indirectly answer that our final error bound can be proved, ii) the intuition of using the spectral mirroring, for which the authors answer in a heuristic manner, iii) why phase preserving, for which the authors blame that other methods do not preserve phases and their accracy iis low, iv) reproduction is not successfult, v) more comparisons with advanced baselines, vi) etc.

**Reviewer Scores:**

I think this paper has two problems, the error seems meaninless, i.e., too loose bound. Developing a loose error bound is rather simple and doable in many cases. The authors use the develroped error bound in the rebuttal messages actively to show the theoretical groundness of their work, however reviewers (including me) seem skeptical on it. They need to show how tight the error bound is and if possible, develop an error bound for each module.

In addition, what if there are noises on phases, not on amplitudes? I think the proposed method cannot address the case. Reviewers may also think in the same way.

Overall, this paper is a boundary case. It has a contribution but at the same time, there exist several apparent enhancements. I recommend that the authors study more on this and improve one more time. I strongly believe that in the next submission, this paper will turn into a very meaning paper in time series forecasting.

---

### Decision · Program_Chairs · 2026-01-26

Reject